# The Relationship between Climate Change, Variability, and Food Security: Understanding the Impacts and Building Resilient Food Systems in West Pokot County, Kenya

Everlyne B. Obwocha [1,*], Joshua J. Ramisch [2], Lalisa Duguma [1] and Levi Orero [1]

1   World Agroforestry (ICRAF), P.O. Box 30766-00100, Nairobi 00100, Kenya;
    l.a.duguma@cgiae.org (L.D.); l.orero@cgiar.org (L.O.)
2   School of International Development and Global Studies, University of Ottawa,
    Ottawa, ON K1N 6N5, Canada; jramisch@uottawa.ca
*   Correspondence: everlynbinyanya@gmail.com

**Abstract:** This study integrated local and scientific knowledge to assess the impacts of climate change and variability on food security in West Pokot County, Kenya from 1980–2012. It characterized rainfall and temperature from 1980–2011 and the phenology of agricultural vegetation, assessed land use and land cover (LULC) changes, and surveyed local knowledge and perceptions of the relationships between climate change and variability, land use decisions, and food (in)security. The 124 respondents were aware of long-term changes in their environment, with 68% strongly believing that climate has become more variable. The majority of the respondents (88%) reported declining rainfall and rising temperatures, with respondents in the lowland areas reporting shortened growing seasons that affected food production. Meteorological data for 1980–2011 confirmed high inter-annual rainfall variability around the mean value of 973.4 mm/yr but with no notable trend. Temperature data showed an increasing trend between 1980 and 2012 with lowlands and highlands showing changes of +1.25 °C and +1.29 °C, respectively. Land use and land cover changes between 1984 and 2010 showed cropland area increased by +4176% (+33,138 ha), while grassland and forest areas declined by –49% (–96,988 ha) and –38% (–65,010 ha), respectively. These area changes illustrate human-mediated responses to the rainfall variability, such as increased stocking after good rainfall years and crop area expansion. The mean Normalized Difference Vegetation Index (NDVI) values ranged from 0.36–0.54 within a year, peaking in May and September. For weather-related planning, respondents relied on radio (64%) and traditional forecasters (26%) as predominant information sources. Supporting continuous climate change monitoring, intensified early warning systems, and disseminating relevant information to farmers could help farmers adopt appropriate adaptation strategies.

**Keywords:** climate change; Kenya; variability; food security; West Pokot

## 1. Introduction

Climate variability and change are a significant threat to food security in Africa and many regions of the developing world, which are largely dependent on rain-fed agriculture [1–4] and, hence, highly sensitive to changes in rainfall patterns. The agriculture-based economies of Africa [5,6] and Asia [7,8] remain reliant on smallholder farming households. Understanding and adapting the knowledge these farmers have of their environments [9,10] to the prediction of the effects of climate change and variability on agricultural systems will be one of the biggest challenges of the current century [11]. The complexity of African agro-ecologies, coupled with a lack of long-term rainfall data from the past century in many African regions, makes it hard to state any conclusions about annual precipitation trends during this time [12]. This has set limits for food production [13], with negative consequences for farmers in terms of their food security and livelihood across the world, especially in developing countries [2,13–15]. Neufeldt et al. [16] explained that

climate change would cause a rise in temperature and change in precipitation patterns, negatively affecting global food production. Such impacts have altered potential crop yield through short-term crop failures and long-term production declines [17], hence, increasing vulnerabilities for smallholder farmers in developing countries [18].

Climate change affects all dimensions of food security (food availability, food accessibility, food utilization, and food system stability), thus, impacting human health, livelihood assets, food production and distribution, and markets [14]. The Food and Agricultural Organization (FAO) defines food security as a situation that exists when people have secure access to sufficient amounts of safe and nutritious food for normal growth, development, and an active and healthy life [14]. However, the major challenge of the 21st century is to achieve food security under marked shifts in climatic risks while using environmentally sound farming practices [19]. The 21st century in particular, is predicted to be the warmest so far [20]. As indicated by Arndt et al. [21], approximately 223 million people are currently undernourished in SSA, and the effects of climate change will increase that number up to 355 million people by 2050.

Like many SSA nations, Kenya has suffered incidences of food deficit. Incidences of prolonged droughts and unexpected changes in normal weather patterns in past years have affected food security, particularly in vulnerable parts of Kenya [22]. This has, in turn, claimed human lives, decimated livestock, and reduced farm output [23,24]. In 2009, over 3.5 million Kenyans faced severe food shortages when failed rainfall seasons led to intense drought [25], which produced a general crop harvest failure [3]. The situation has worsened, and the frequency with which the country experiences cyclical food crises has reduced from 20 years (1964–1984) to 12 years (1984–1996), to 2 years (2004–2006), to yearly (2007/2008/2009/2010/2011/2012) [26]. Wakibi et al. [27] show that 30 percent of Kenyan households are food insecure, which means they do not have access to sufficient food to sustain an active and, healthy life for all household members. USAID [24] has also indicated that approximately one-third of Kenya's population is food insecure. According to the Food Security Index of the World Food Program, a report by Ng'ang'a [28] listed Kenya among the 15 countries prone to food insecurity. Despite these uncertainties, Kenya should aim to achieve food security and end hunger in the face of the ongoing impacts of climate change and variability as it is core to the Sustainable Development Goals.

West Pokot County is one of the food deficient and food insecure counties in Kenya [23]. As a semi-arid region, the county experiences a highly variable climate, such as drought and unpredictable rainfall patterns, but existing research on the resulting food insecurity and related humanitarian disasters has largely treated local populations as victims rather than agents of potential change (e.g., [23]). Most previous studies in Kenya indicate that the impact of climate change on crop and livestock productivity has ambiguous and complex impacts on food security that require local parameterisation [29,30]. Studies have established that undertaking climate change impact assessments on a local scale is essential as it allows the exploration of local agronomic management practices and their incorporation into adaptation strategies formulation [31,32]. Indeed, given an existing context of substantial variability and uncertainty in most Kenyan production systems (e.g., of rainfall, temperature), it is increasingly apparent that adaptation measures will not be adopted without building upon how local people perceive and respond to long-term processes of climate change [29]. This study addresses this gap by using West Pokot as a case study to demonstrate how the ambiguous temperature and rainfall data for the county nevertheless account for clear land use and land cover changes, which can be meaningfully interpreted using the local population's understandings of climate and food security outcomes. Integrating these complex knowledge systems is essential for designing and implementing successful adaptation interventions [33,34] and for climate risk management [35–37].

Remote sensing data provides largely unexploited opportunities in agriculture to assess land use and crop growth on parcel, farm, and regional scales [38]. Land use and land cover (LULC) change is, therefore, central to food security assessments and this

study contributes to understanding the role of land use in the sustainability of global food systems amid climate change effects. According to Darkoh [39], the rapidly accelerating change in the landscape is associated with a wide variety of issues, including declining biodiversity, global climate change, food security, and land degradation, which applies to soils, vegetation, and water depletion. The Normalized Difference Vegetation Index (NDVI) variations in both space and time scales that can be assessed remotely, are important for illustrating vegetation–climate feedback mechanisms at varying crop stages, which can assist policy makers with proactive and reactive risk measurements [40–43]. This is especially relevant for West Pokot County where systematic data of such relationships is lacking. This sort of data can be applied for the identification of crop species and soil management practices.

With this study, we sought to answer the following questions: How has rainfall and temperature in West Pokot changed from 1980–2011? What LULC changes can be observed in the study area between 1984–2010? What is the phenology of agricultural vegetation in West Pokot County? Do households associate food shortage with climate change and variability? The study provides a comprehensive knowledge base in use of satellite sensor-based maps and statistics that can be used to develop strategies for agricultural and agro-pastoral livelihoods. It helps in the understanding of what location-specific policies and strategies can be developed in the area for land use planning, natural resource management, and adaptation in the face of climate change. This will offer the West Pokot people a more sustainable and desirable pathway to food security.

## 2. Data and Methods

### 2.1. Site Selection

The study sites were selected to reflect the land classification typology used in West Pokot. The resident Pokot ethnic group themselves utilize and classify their land on the basis of altitude, rainfall, and agricultural potential. The study used this typology to identify three zones of differing agroecological potential (Table 1), which guided selection of three sites (at the division level) within the County (Figure 1). Kapenguria division was chosen to represent the highland zone (known as "masop"), which receives the highest amount of annual precipitation. Medium potential areas ("kamas") were defined as those adjacent to the highlands and were represented by Chepareria division. Most of the areas in Pokot South sub county are in this category. Finally, low potential, mostly arid areas ("tow") far away from the highlands were represented by Kacheliba division. Most of the areas in North Pokot sub-county belong to this category. Table 1 summarizes the study area description and Figure 1 depicts the study sites.

**Table 1.** Study area description.

| Study Areas | Area (Square kms) | Agricultural Potential | Pokot Land Classification Type | Average Annual Rainfall (mm) | Average Annual Temperature ($^{\circ}$C) |
|---|---|---|---|---|---|
| Kapenguria | 335.6 | Highland | Masop | 1600 | <21 |
| Chepareria | 495 | Midland | Kamas | 600 | 24 |
| Kacheliba | 925.4 | Lowland | Tow | 300–400 | 28 |

At the time of the study, Counties had replaced Districts as the largest subnational administrative unit in Kenya, under the 2010 Constitution. Sampling however relied on the pre-2010 administrative units (divisions, locations, sub-locations, and villages), since documentation had yet to be updated at the local level. Sources: [44–46]

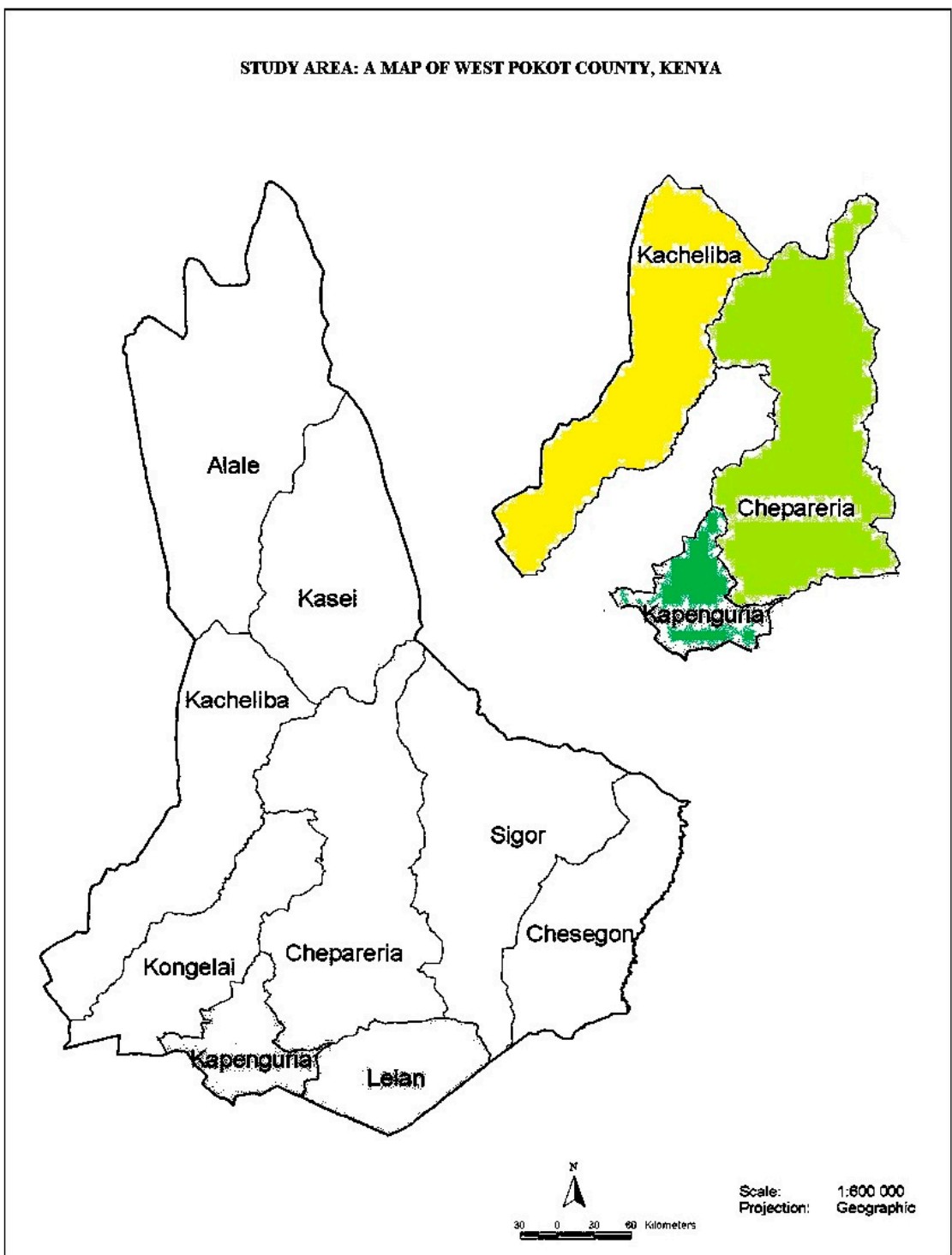

**Figure 1.** Study sites (redrawn from FAO 2006).

*2.2. Sample Size and Sampling Procedure*

The target population was comprised of households from the three selected divisions (Kapenguria, Chepareria and Kacheliba) of West Pokot County. According to KNBS (2010), West Pokot had a population of 512,690 people and 93,777 households. A sample of 124 households was selected to form the study population using Nasuirma Model formula expressed as follows:

$$n = \{NC_v{}^2\}/\{C_v{}^2 + (N-1)\epsilon^2\}$$

where:

*n*—the sample size
N—the target population (93,777)
**C$_V$**—the coefficient of variation (0.5)
$\epsilon$—the tolerance of desired level of confidence at 95% level (0.05)
$(93{,}777 \times 0.5^2)/(0.5^2 + (93{,}777 - 1)\, 0.05^2) = 23{,}444/234.69 = 100$ respondents

The sample size is often increased by 30% to compensate for nonresponse [47].

A multistage random sampling was adopted to select participating villages and households for interviews. Kapenguria, Chepareria and Kacheliba divisions were purposively sampled based on food security potential, geographical location, and vulnerability to climatic change and variability. The locations within the three divisions were listed and categorized on the basis of the land use system activities, accessibility, and the extent to which they were perceived to be prone to climate variability and change. Twenty (20) locations from the divisions were randomly selected and respective sub locations and villages were listed along each location. The participating villages were randomly selected and the number of households for the study was obtained. The questionnaires were administered in the three divisions proportional to their share of the total population, whereby 36 respondents from Chepareria, 56 from Kapenguria and 32 respondents from Kacheliba were randomly selected for interviews. The age of the respondents varied from 20 to 80 years and the majority (51%) had primary education, with 13% having no formal education. In addition to the random sampling of rural respondents, purposive sampling was used to select 20 key informants from relevant institutions, including Water and Resource Management Authority, Ministry of Agriculture, Kerio Valley Development Authority and community leaders. The interviews followed a semi structured format, with an in-depth focus in regard to the area of expertise of the informant.

### 2.3. Data Acquisition

The survey was conducted in Jan-March 2013, before the long rains season and used both structured and semi structured questionnaires to elicit information for ground truthing, verification, providing basic information and for the study of perceptions. As noted above, the multistage sampling technique selected 124 household respondents, while purposive sampling identified 20 key informants. Data on rainfall and temperature for West Pokot County was obtained from the Kenya Meteorological Department.

Satellite imagery with 30 m spatial resolution from Landsat 5 and Landsat 7 was used to analyze LULC changes in West Pokot County. The spatial resolution of Landsat imagery and its multispectral characteristics make it a suitable source of data for environmental and climate studies since various band combinations provide information on the land surface and its properties. SPOT 5 multispectral data sets for the years 1984, 1990, 2000 and 2010 were used for NDVI analysis.

### 2.4. Estimation Strategy

Logistic regression was applied to test the hypothesis that the probability of taking measures to reduce exposure to climate change was related to several predictor variables [48]. Our binary regression equation was of the form:

$$\text{logit}(Y) = \beta_0 + \beta_1 X_1 + \beta_2 X_2 + \ldots + \beta_n X_n + \varepsilon$$

Stepwise regression began with 71 possible predictor variables for the dependent variable: "Do you take any measures (in previous and current year) to reduce your exposure to the impacts of climate change on food security risk?" The first reduced model had 18 predictor variables. We conducted a second stepwise regression to get a second reduced model, considering that we only had 124 observations and 18 was a case overfitting.

### 2.5. Image Processing

For each of the four sets of geo-referenced Landsat data, band combinations generated color composites to allow for interpretation. Layer stacking was done in ERDAS Imagine 15® software [49], using the relevant bands (i.e., excluding the Coastal/Aerosol, Panchromatic, Cirrus, and Thermal Infrared bands). Multi-band (multispectral) images covering the entire county were obtained, using bands 2, 3, 4, 5, and 7 (Landsat 5), and bands 1, 2, 3, 4, 5, and 7 for the more recent Landsat 7 images.

The bands help with vegetation enhancement and color contrast. The Normalized Difference Vegetation Index (NDVI) was extracted for each year of study and maps of NDVI were generated. Multispectral vegetation indices were represented by the algebraic combination of remotely sensed spectral bands that indicated the phenology of the vegetation cover. For the state of the crops, the different sensitivity of the mentioned electromagnetic spectra was used to estimate the productivity of the study area. Software used included: ERDAS Imagine 2015® [49] to perform the image processing and enhancement, and analyze multispectral image data by means of qualitative and quantitative approaches, Arc GIS(™) Version 10.5 [50] for developing maps, and ERDAS Imagine MosaicPro tool for the mosaicking.

### 2.6. Data Analysis

Image classification was examined using visual analysis, classification accuracy, band correlation, and decision boundary. The study considered the requirements set out in the 2006 Intergovernmental Panel on Climate Change (IPCC) [51] guidelines, which define the seven broad land use classes that countries are required to report on under the United Nations Framework Convention on Climate Change (UNFCCC) as: Forestland, Cropland, Open Grassland, Wooded Grassland, Wetland/Open Waters, Settlements, and Other land.

NDVI was used to identify vegetated areas and their associated health. The NDVI anomalies for the years and changes in productivity observed between the subsequent years were also interpreted. The GIS approach provided a spatial framework to support spatio-temporal analysis of Landsat data. The GIS geo-processing tools analyzed information based on vegetation indices and other spatial data.

The variability and time series trend characteristics of rainfall and temperature data were analyzed to inspect the changes in the historical period. Trend detection and analysis were achieved through time series decomposition for both datasets. Multiple logistic regression was used to analyze the adaptive measures taken by respondents in response to climate change. These analyses and graphing were conducted in R software (version 4.1.0) [52] and several of its packages [53–59].

### 3. Results

#### 3.1. Characterizing Annual and Seasonal Rainfall and Temperature from 1980–2011 in West Pokot County

3.1.1. Rainfall Trend Analysis

The analysis of monthly rainfall variation for the period between 1980 and 2011 (Figure 2) indicated that rainfall is unpredictable and unreliable, and does not coincide with cropping seasons. As such, it does not provide sufficient water long enough for crop cultivation.

The long-term trend analysis in annual rainfall showed no significant change over time at the County level (Figure 2). Rainfall amounts for West Pokot County fluctuated between a maximum of 1347.9 mm in 1982 and minimum of 619.4 mm in the year 2000 with an average annual rainfall of 973.4 mm (Figure 2). Twelve years (1982, 1988, 1992, 1993, 1994, 1996, 1997, 1999, 2001, 2006, 2007 and 2008) experienced above average precipitation, while seventeen years (1980, 1981, 1983, 1984, 1985, 1986, 1987, 1989, 1990, 1991, 1995, 2000, 2002, 2003, 2004, 2005 and 2009) recorded below average precipitation.

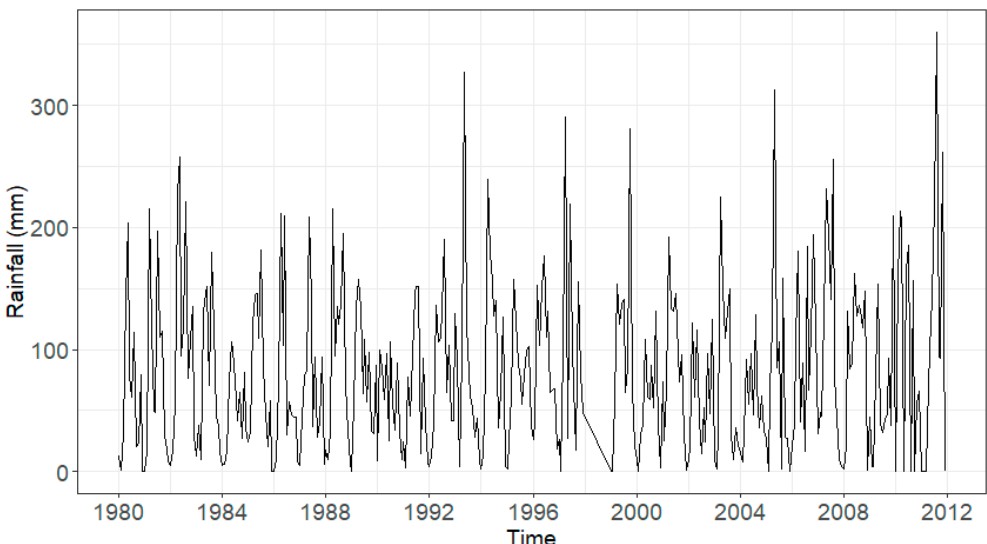

**Figure 2.** Annual rainfall trend in West Pokot County (1980–2011).

The years that experienced the highest amount of rainfall were 1982 (1347.9 mm) followed by 2011 (1307.5 mm), 2007 (1210.7 mm) and 2010 (1209.6 mm). The years which recorded the least rainfall amount were 2000 (619.4 mm) and the year 1984 (631.61 mm). This confirmed reports from the respondents that rainfall had become irregular and unpredictable in terms of amount and distribution affecting crops, such as maize and beans especially during the maturity period. "The year 2000 was a year I cannot forget, I slept hungry most of the days due to crop failure, there were no rains for my crops to grow" a respondent lamented. After the 2000 drought, annual rainfall increased, but with considerable variations further jeopardizing food security.

There was no obvious increasing or decreasing trend in the rainfall data over the study period (Figure 3). The trend component was irregular. Respondents did, however, believe that rainfall amount was diminishing over time, which is not supported by these data. Rainfall levels are not found to significantly decline during the study period, rather a trend in variability in rainfall distribution was. However, the data suggested the presence of seasonality, as expected, where rainfall is heavy or less in specific months of the year. Further examination revealed that rainfall peaked in April, while January recorded the lowest rainfall.

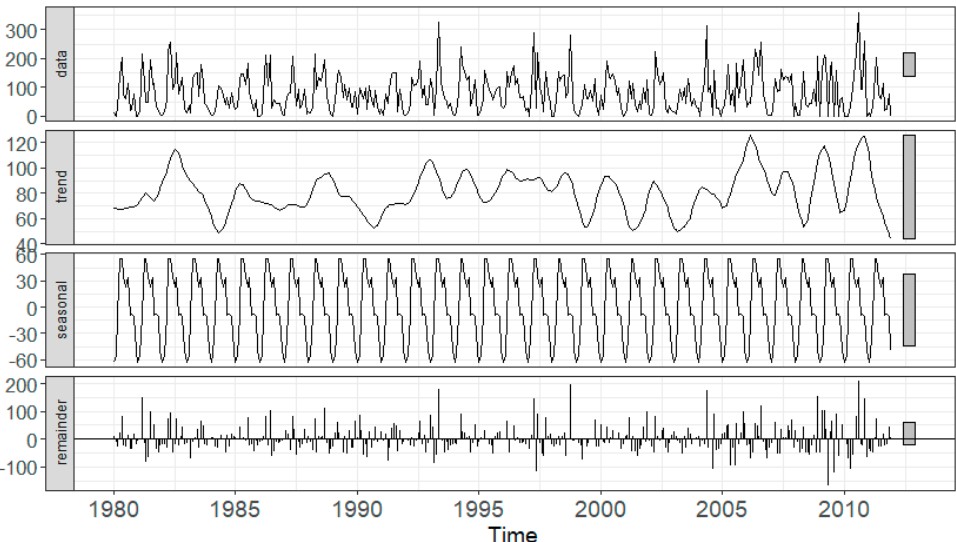

**Figure 3.** Decomposed time series of rainfall in West Pokot County (1980–2011).

### 3.1.2. Temperature Trend Analysis

Lowland West Pokot: Average daily temperature per year was calculated for the lowlands (Figure 4).

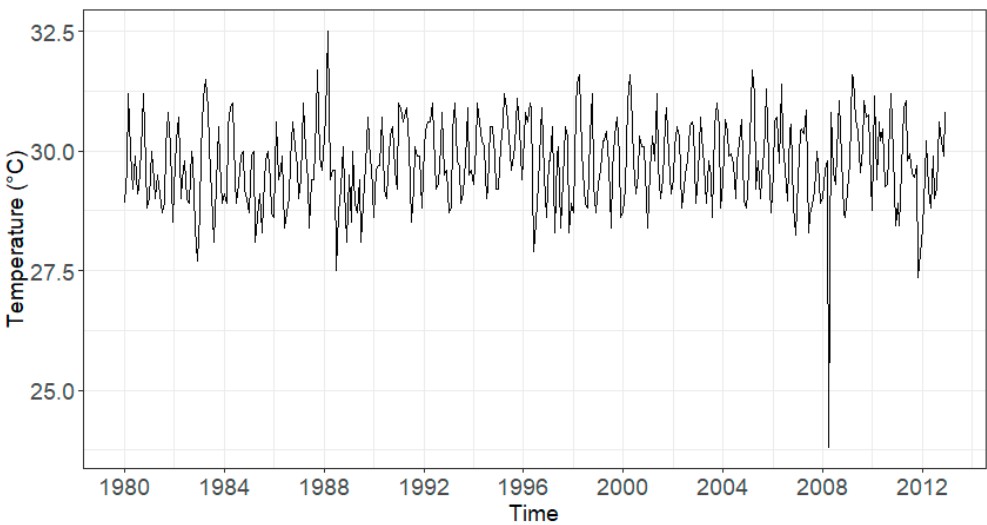

**Figure 4.** Average daily temperature per month in West Pokot County lowlands (1980–2012).

The average annual temperature for the lowlands over the study period 1980–2012 was 29.74 °C. While annual average results showed that the year 1985 recorded the lowlands' lowest annual daily temperature (29.13 °C), monthly averages showed that April 2008 had the lowest average temperature (23.8 °C). We retained the lowest temperature in April 2008 in further analyses, even though it was an outlier. On the other hand, whereas the highest annual average temperature was in 2009 (30.38 °C), the highest monthly average temperature was in March 1988 (32.5 °C). The lowlands' mean annual maximum temperature showed warming trends over the study period, with considerable interannual variations, apart from 1997–2006 where the variations were minimal (Figure 5). Overall, there was a significant warming trend in the lowlands' average temperature, which rose +1.25 °C over the study period.

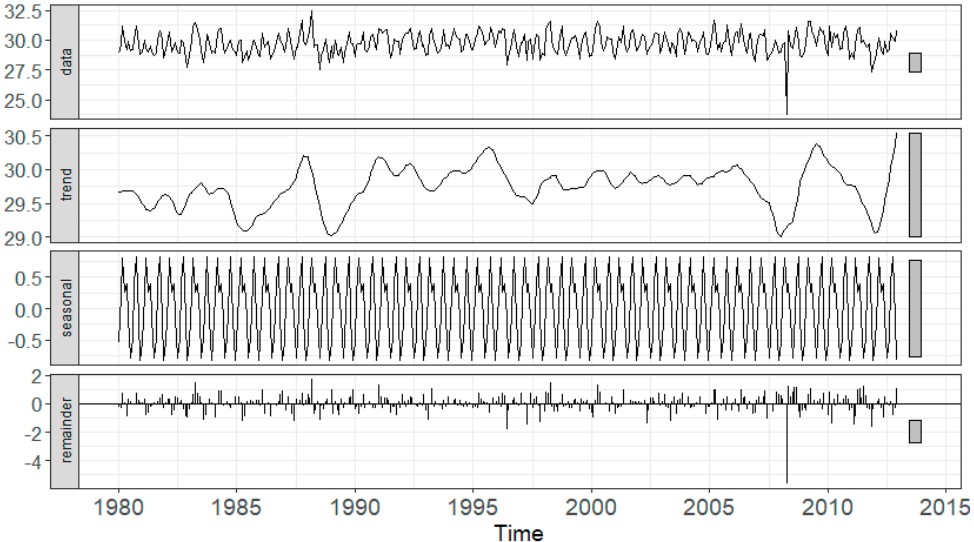

**Figure 5.** Decomposed time series of average temperature in lowland West Pokot County (1980–2012).

The decomposed time series showed the presence of seasonality in the data of lowland West Pokot. Further examination reveals that average temperatures peaked in March and

October, while July recorded the lowest temperatures. The trend analysis showed a steady increase between 1997 and 2006, which became irregular irregular in the years following.

Highland West Pokot: Average daily temperature per year was calculated for the highlands (Figure 6).

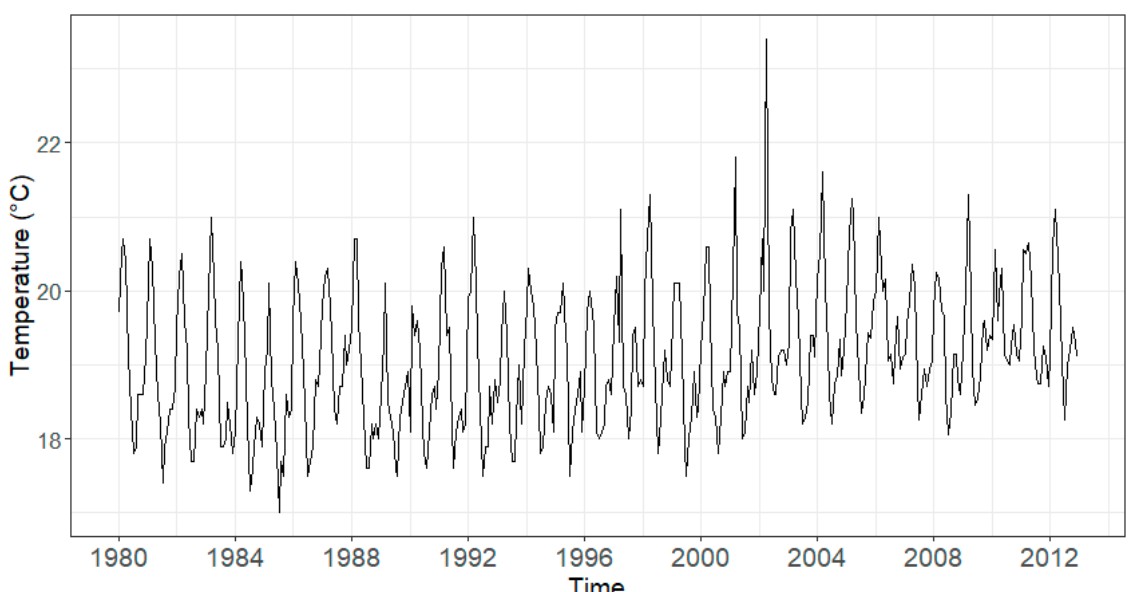

**Figure 6.** Average daily temperature per month in West Pokot County highlands (1980–2012).

The time series showed an increasing trend in the average temperature by month since 1980. We then decomposed the time series to isolate its components as in Figure 7.

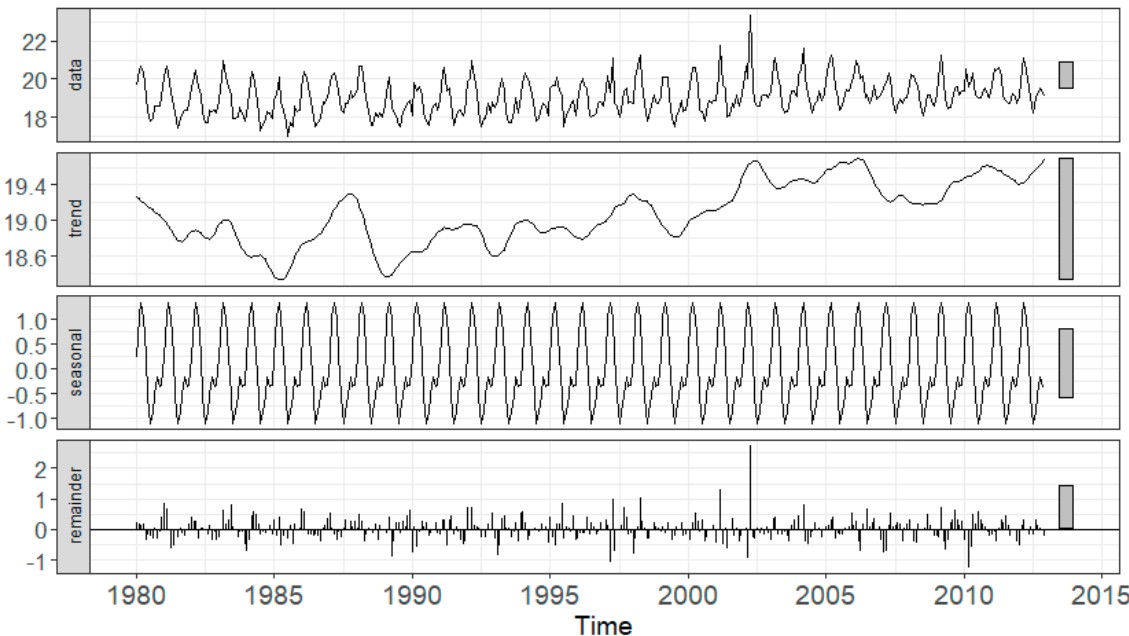

**Figure 7.** Decomposed time series of average temperature in highland West Pokot County (1980–2012).

The data showed the presence of seasonality in the temperature data. Further examination reveals that average temperatures peaked in March, while July recorded the lowest temperatures. Furthermore, unlike the lowland areas, there was a trend in the time series showing that average temperatures had increased since 1990 in the highland areas. In the highland areas, the year 1985 recorded both the lowest annual daily temperature

(18.39 °C) and lowest monthly average temperature in July 1985 (17 °C). On the other hand, whereas the highest annual average was 19.68 °C in 2005, the highest ever monthly average temperature was in April 2002 at 23.4 °C.

### 3.2. Land Use Land Cover (LULC) Changes

A total of seven land cover classes were considered namely: Forestland, Cropland, Open Grassland, Wooded Grassland, Wetlands/Open Waters, Settlements, and Other land. Figure 8a–d for 1984, 1990, 2000, and 2010, respectively, show maps of the interpreted images of West Pokot County.

In 1984, wooded grassland was the most dominant LULC in the landscape covering 533461 ha of the total studied area, followed by open grassland (196,775 ha), forestland (169,452 ha), otherlandSs (32,058 ha), open water (1139), cropland (794 ha), and settlements (32 ha) (Table 2). By 2010 there was a substantial increase in croplands (+4176%). During the same period, other lands (bare lands and rocky areas), open grasslands, and forestland declined by −71%, −49%, and −38% respectively. The greatest expansion of cropland occurred between 2000 and 2010 (from 8254 ha to 33,931 ha), whereas forestland recorded its lowest area (86,460 ha) in 2000 followed by a modest increase in the year 2010. However, area under wooded grassland, open water, and settlement have consistently increased by +28%, +247%, and +266% respectively. There was a decline in area coverage for open water in 1990, however it increased considerably in 2000–2010.

**Table 2.** Changes in area by land use types in West Pokot County (1984–2010).

| Landuse/Area in Hectares (Ha) | 1984 | 1990 | 2000 | 2010 | Net Change |
|---|---|---|---|---|---|
| Cropland | 794 | 4938 | 8254 | 33,931 | +33,138 (+4176%) |
| Forestland | 169,452 | 122,004 | 86,460 | 104,443 | −65,010 (−38%) |
| Open Grassland | 196,775 | 123,176 | 110,502 | 99,787 | −96,988 (−49%) |
| Open Water | 1139 | 498 | 1070 | 3957 | +2818 (+247%) |
| Otherland | 32,058 | 9910 | 8978 | 9361 | −22,698 (-71%) |
| Settlements | 39 | 47 | 70 | 142 | +103 (+266%) |
| Wooded Grassland | 533,461 | 673,114 | 718,359 | 682,734 | 149,274 (+28%) |

Accuracy assessment of the classification was computed after field verification and validation. It compared the classified image to ground truthed data collected from the field. A set of random points from the ground truth data was created and compared to the classified data in a confusion matrix. Table 3 shows the accuracy assessment results. The overall accuracy of the classification is 74.42% with a Kappa coefficient of 0.674. Relative to the size of the region covered by the study, the classification accuracy is relatively good.

**Table 3.** West Pokot County (2010 landcover map error matrix).

| Class EVF | Settlement | Forestland | Cropland | Otherland | Open Grassland | Wetland | Wooded Grassland | Total |
|---|---|---|---|---|---|---|---|---|
| Settlement | 1 | 0 | 1 | 0 | 0 | 0 | 0 | 2 |
| Forestland | 0 | 8 | 0 | 0 | 0 | 0 | 0 | 8 |
| Cropland | 1 | 0 | 12 | 0 | 3 | 0 | 3 | 19 |
| Otherland | 0 | 1 | 0 | 5 | 0 | 0 | 0 | 6 |
| Open grassland | 0 | 0 | 0 | 0 | 2 | 0 | 0 | 2 |
| Wetland | 0 | 0 | 0 | 0 | 0 | 1 | 0 | 1 |
| Wooded grassland | 0 | 0 | 0 | 0 | 2 | 0 | 3 | 5 |
| Total Sampled Points | 2 | 9 | 13 | 5 | 7 | 1 | 6 | 43 |



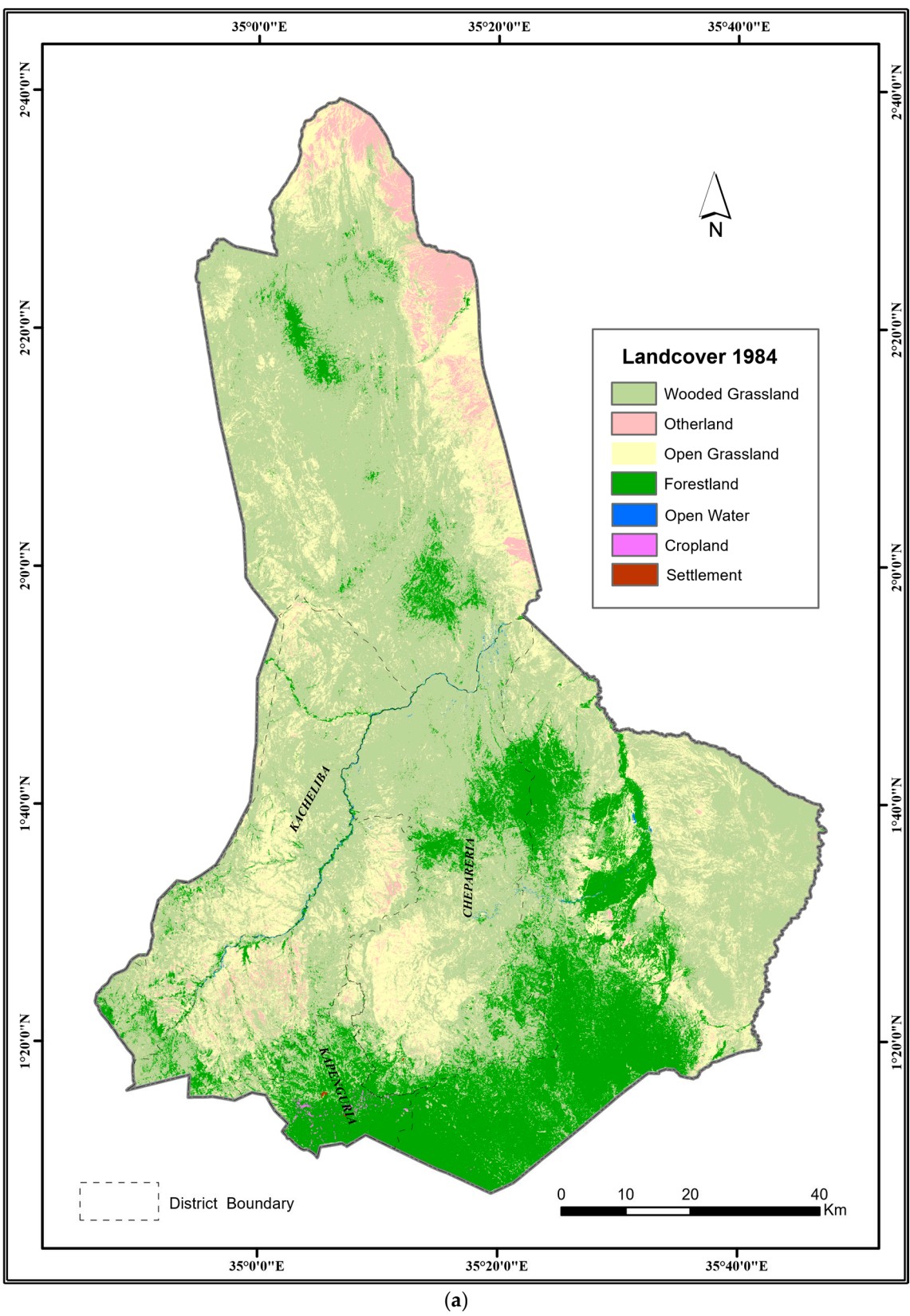

(**a**)

**Figure 8.** *Cont.*

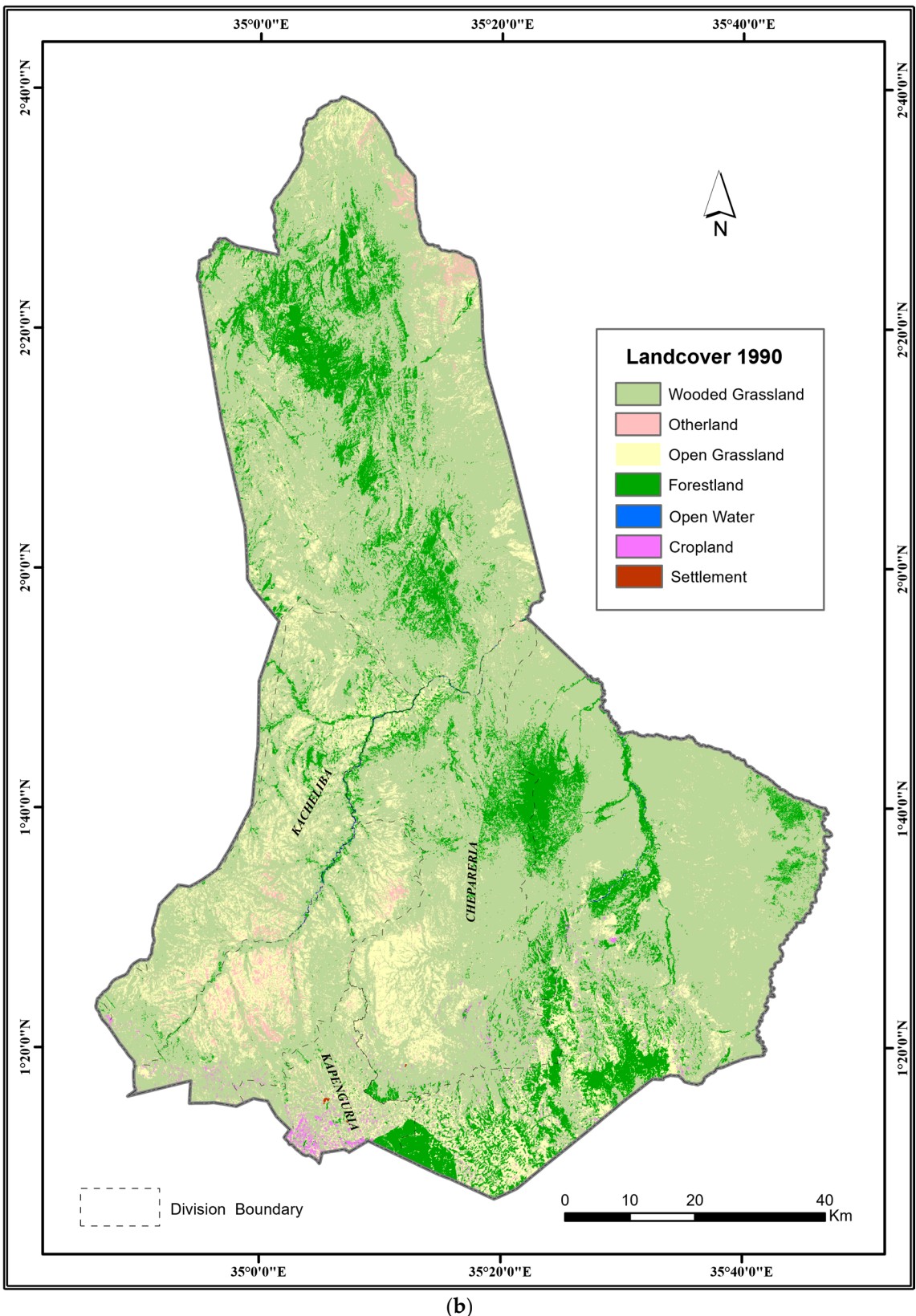

(**b**)

**Figure 8.** *Cont.*

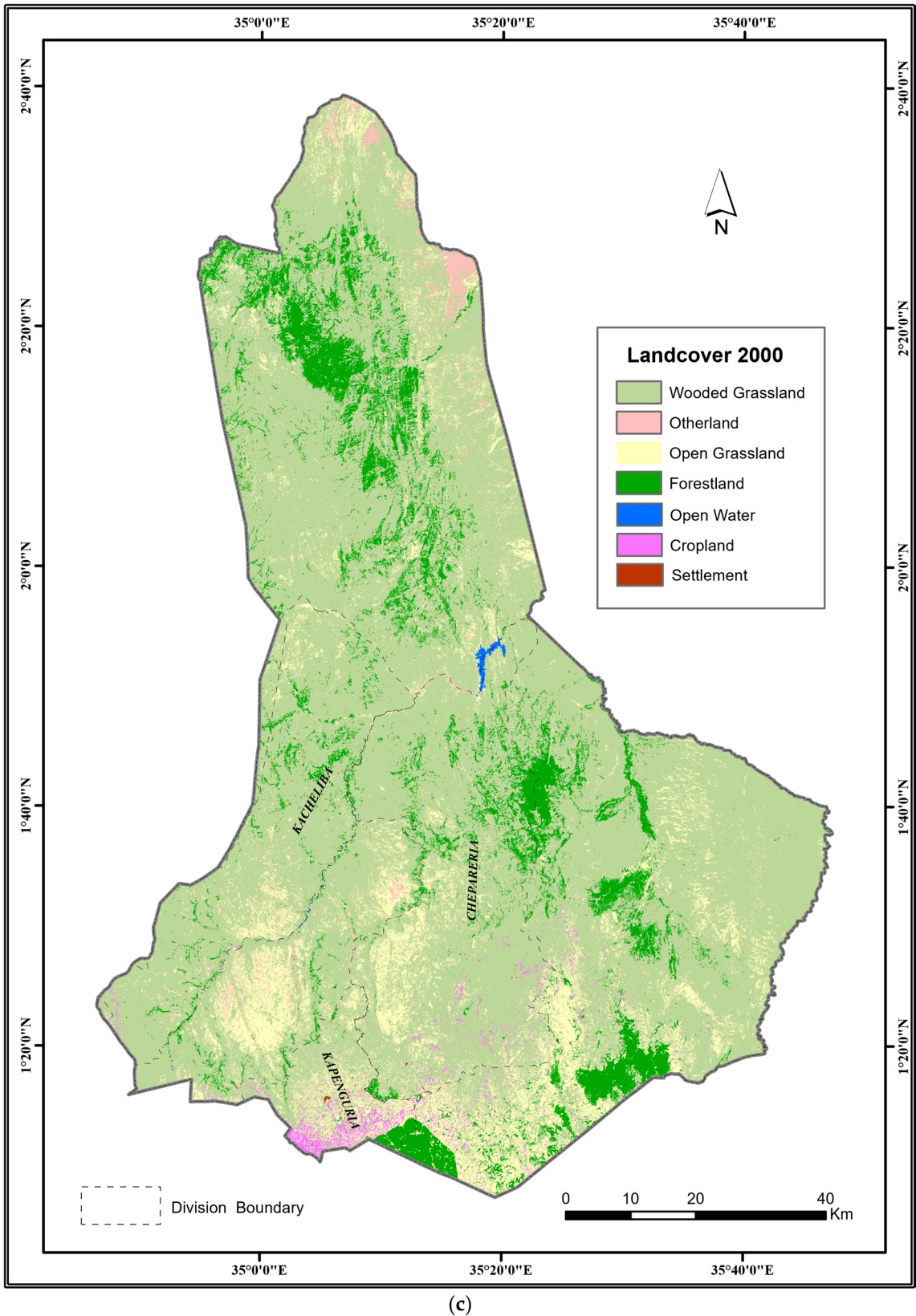

(**c**)

**Figure 8.** *Cont.*

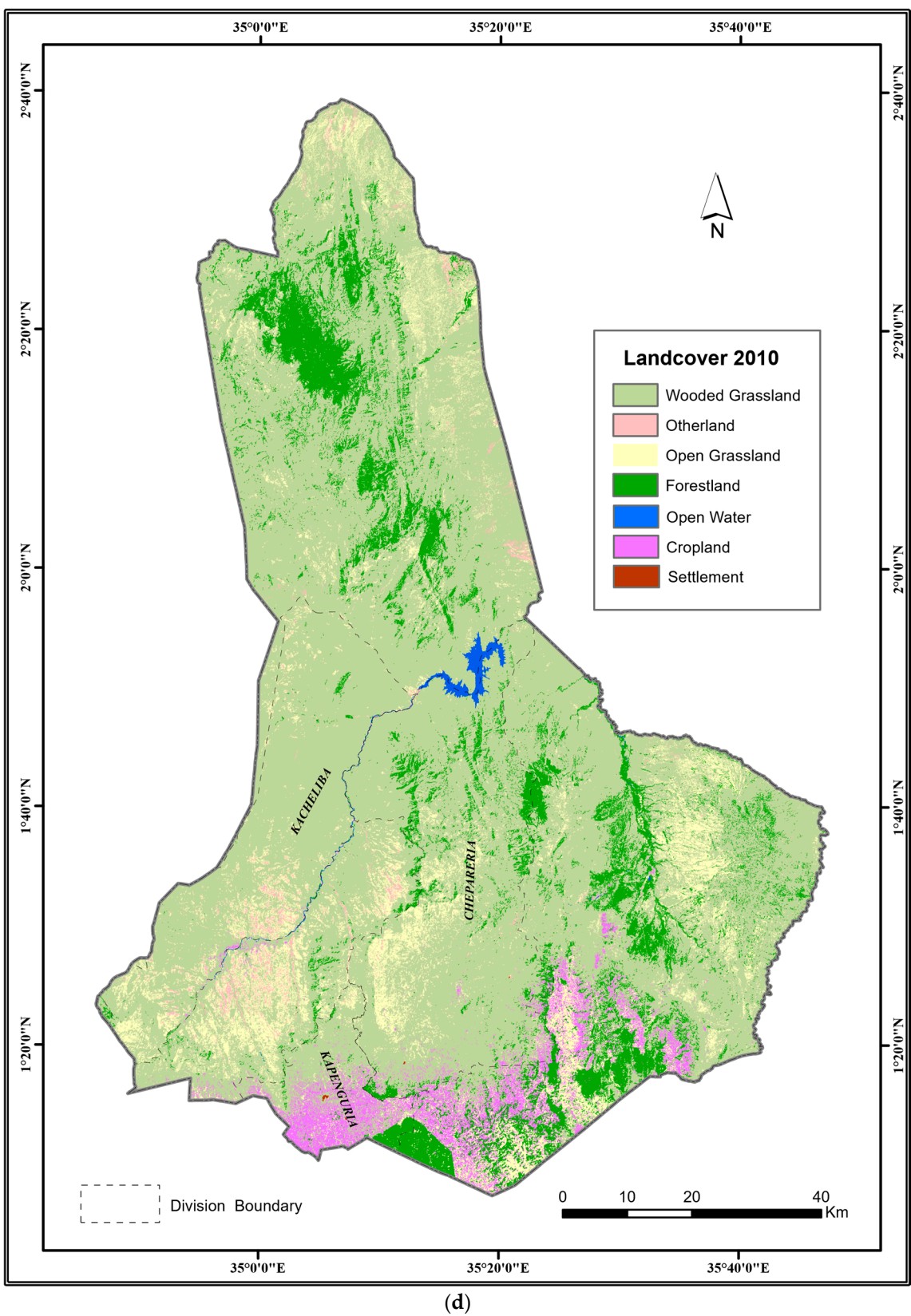

(**d**)

**Figure 8.** (**a**) Land use land cover, 1984; (**b**) Land use land cover, 1990; (**c**) Land use land cover, 2000; (**d**) Land use land cover, 2010.

### 3.3. The Phenology of Agricultural Vegetation in West Pokot County

The inter annual variation in NDVI was assessed for West Pokot County for the period from 1984 to 2010 and the vegetation production represented by NDVI images, Figure 9a–d.

There was a high variability of vegetation cover in West Pokot. The normal vegetation pattern has been disturbed and dominated by significantly greater variation from one year to another. Normal vegetation meant healthy vegetation while poor meant moisture-stressed vegetation.

The county was dominated with healthy and some patches of moisture-stressed vegetation in 1984 (Figure 9a), even though the amount of rainfall it recorded was low (631.6 mm). This followed the above average rainfall amount (1347.9 mm) experienced in 1982 which had enhanced adequate soil moisture that supported vegetation growth in the year 1982 followed by slightly below average rains (956.8 mm) in 1983 which sustained the growth.

The year 1990 showed a poor and very poor state of vegetation especially in some parts of the midlands and lowlands (Figure 9b). This follows a continued stretch of below average rains (1984, 1985, 1986, 1987 and 1989). Even though 1988 recorded above average rains (1069 mm), the county was recovering from frequent dry spells and still suffering from moisture stress, hence, negligible impacts on vegetation. This is especially evident in the lowlands, where poor vegetation dominated in 1990. Chepareria was the most affected part of the midlands. The highlands, especially the Kapenguria area, also recorded poor and only a few patches of good vegetation. Only along the River Suam was vegetation in a normal state.

The green cover improved for the year 2010 (Figure 9d) following the high rains (1209.6 mm) recorded that year, and good rains experienced in the previous years (2005–2008) meaning the vegetation was healthy and received enough rain. The highlands indicated very good vegetation and the midlands were dominated by normal vegetation. Some parts of the lowlands however recorded poor vegetation.

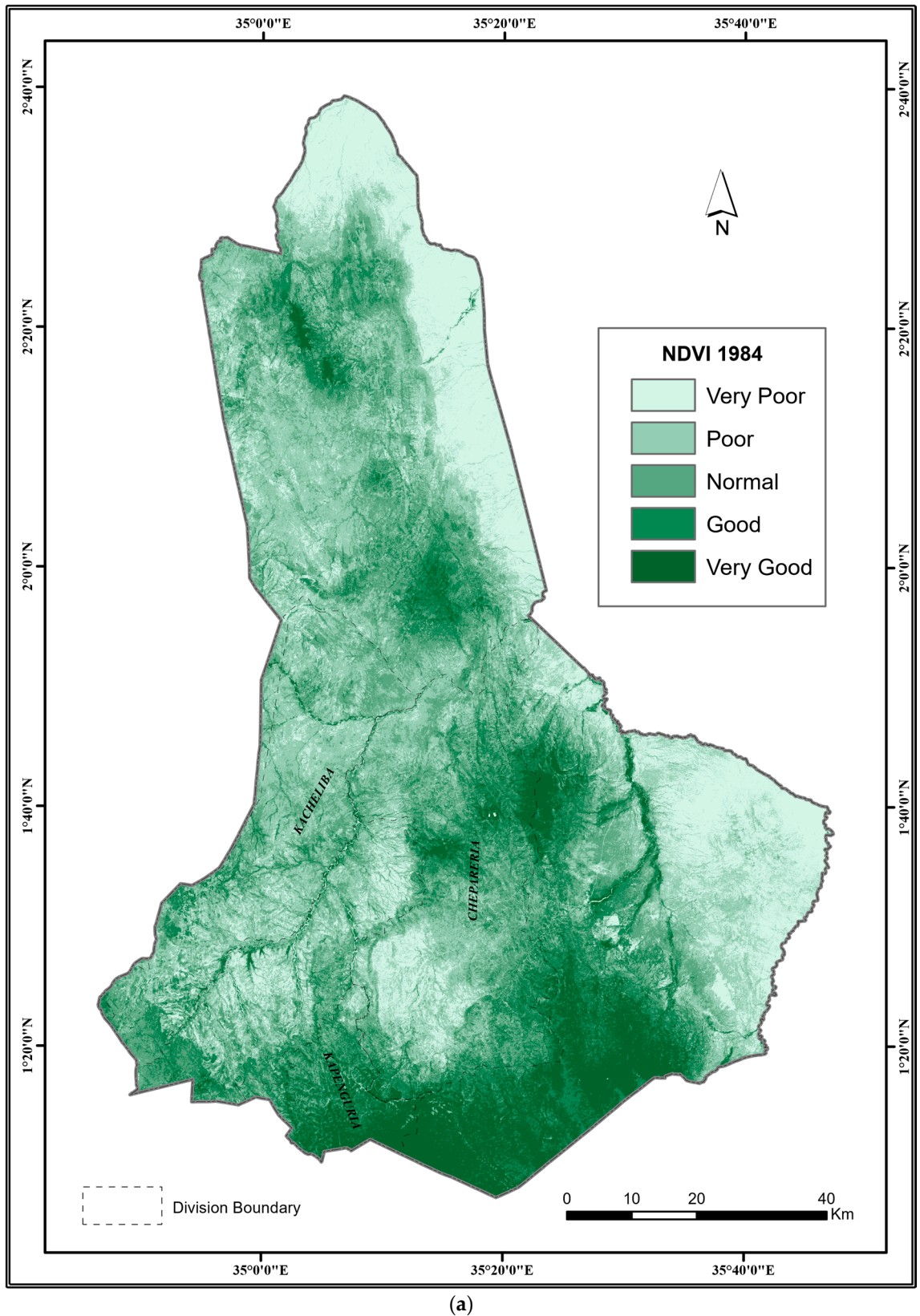

(**a**)

**Figure 9.** *Cont*.

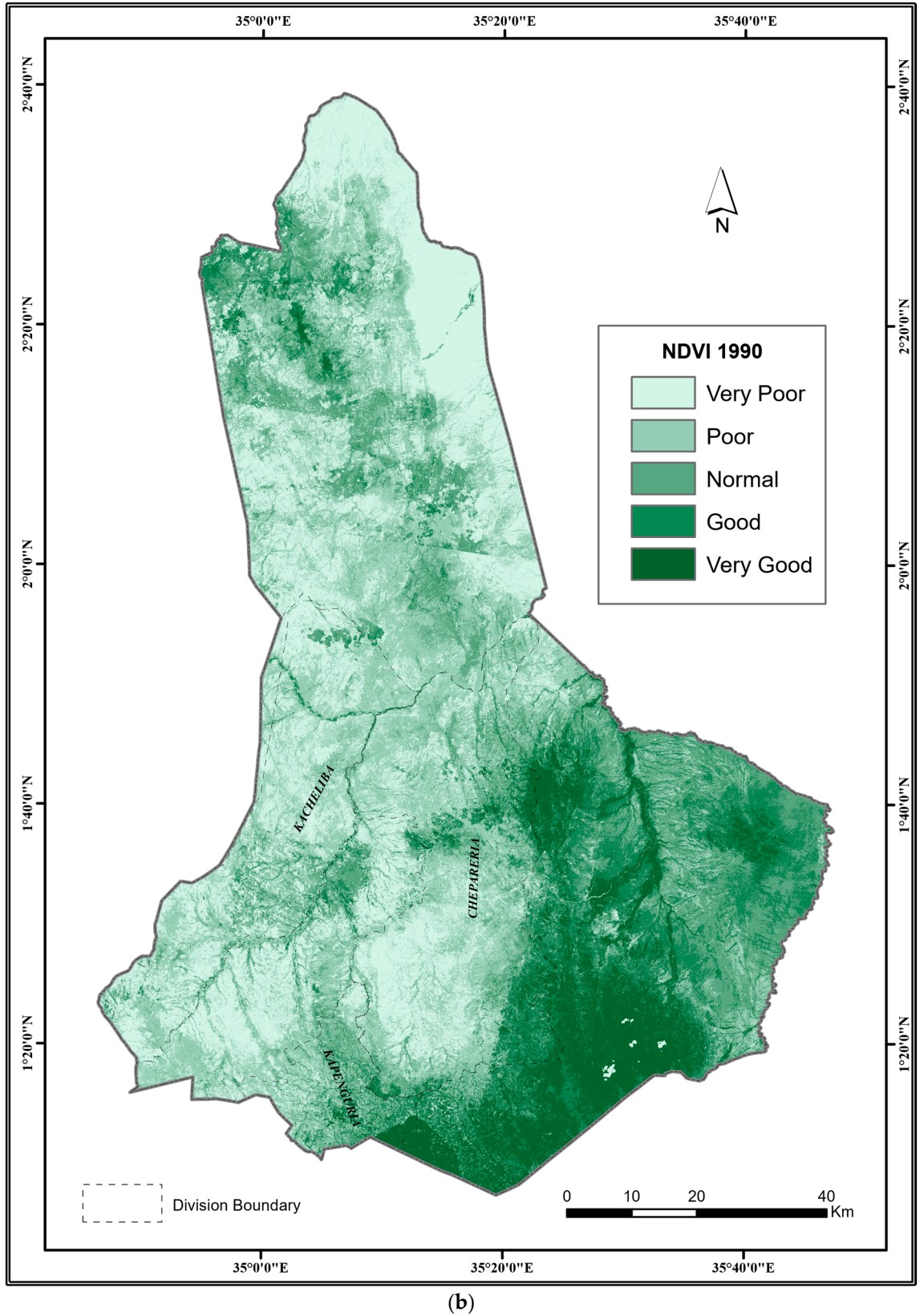

(**b**)

**Figure 9.** *Cont.*

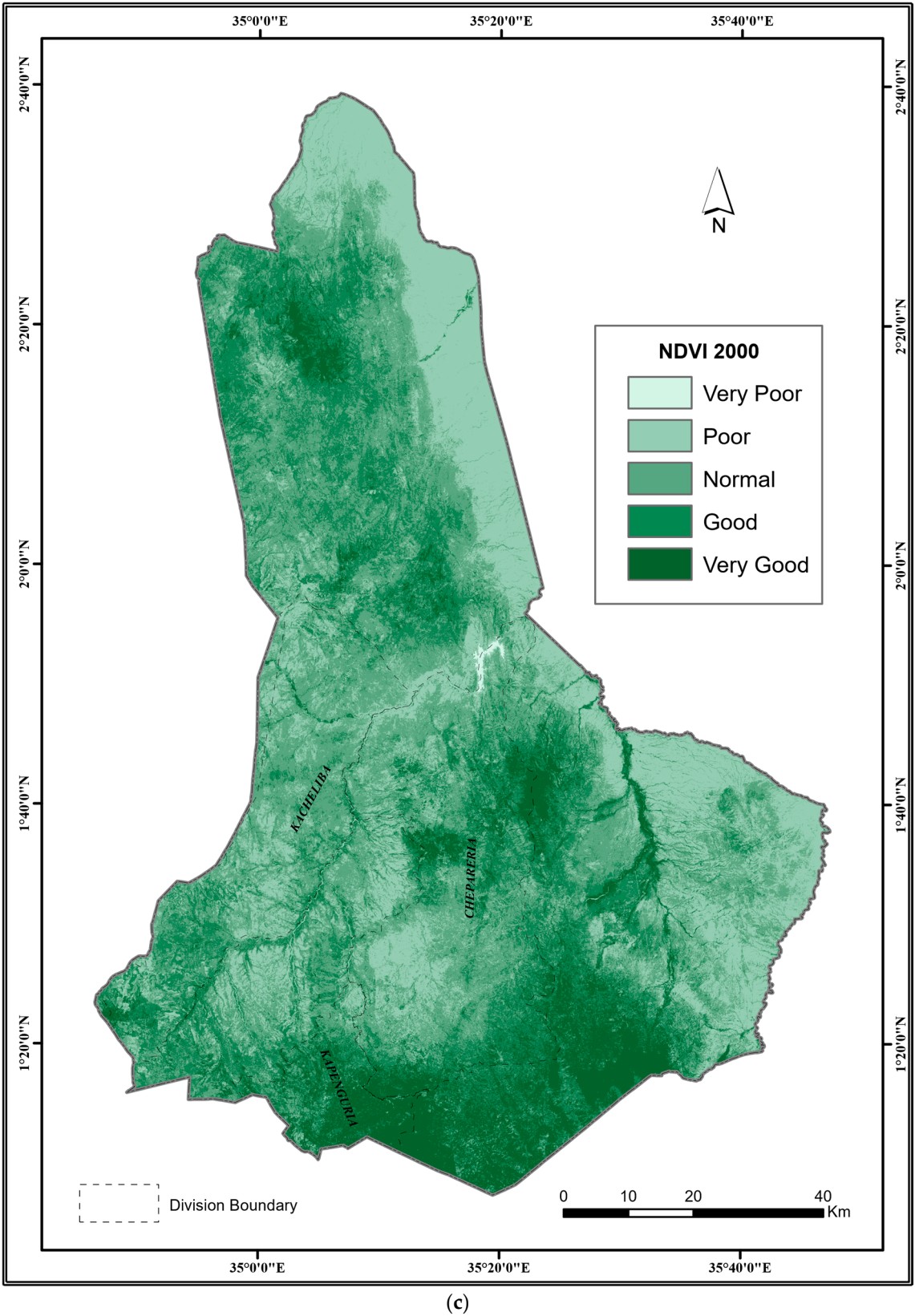

(**c**)

**Figure 9.** *Cont.*

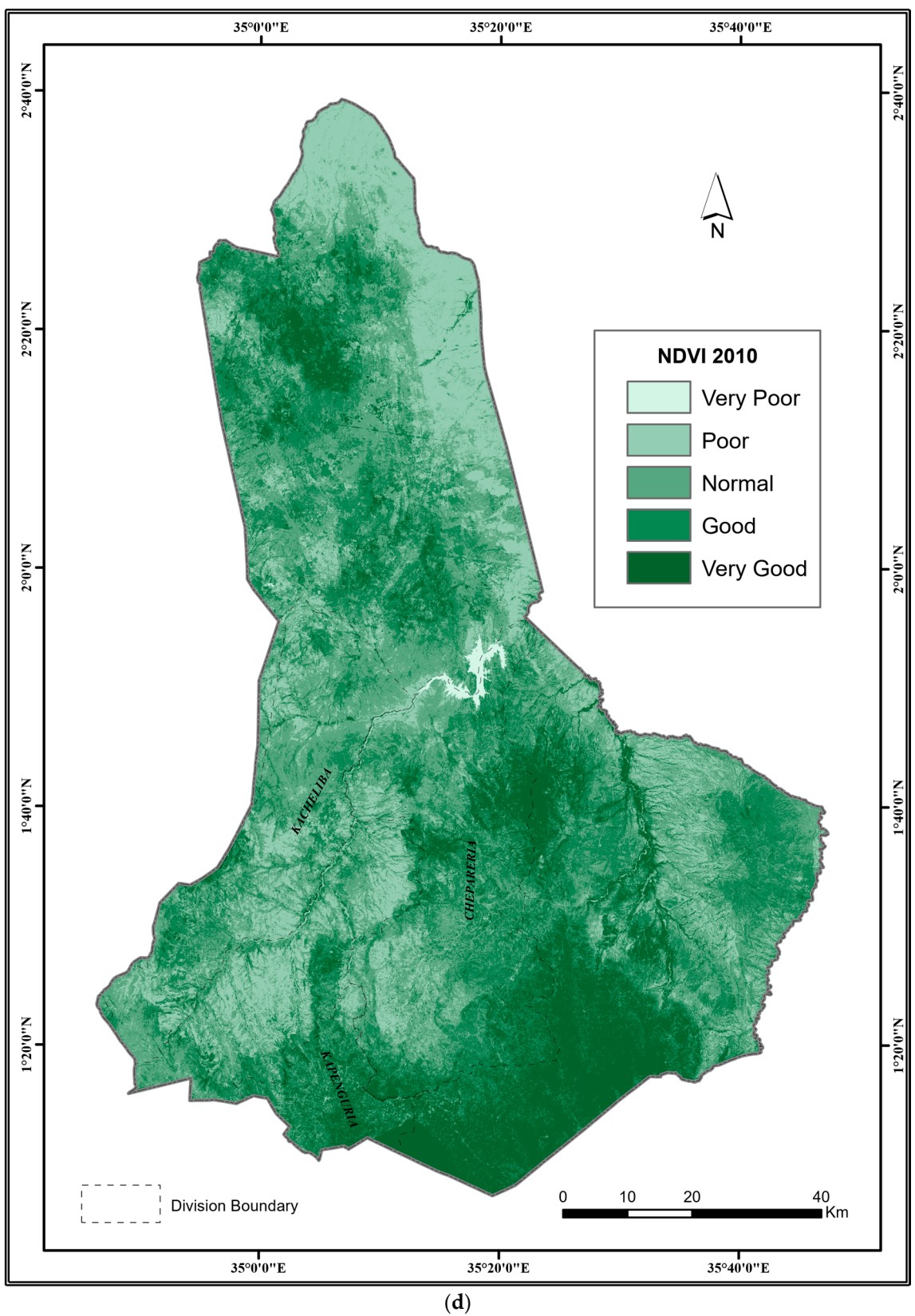

**Figure 9.** (**a**) Vegetation production, 1984; (**b**) Vegetation production, 1990; (**c**) Vegetation production, 2000; (**d**) Vegetation production, 2010.

Analysis of NDVI Anomalies in West Pokot County

The NDVI time series in West Pokot County show very low values from January to February. The NDVI values then increase and peak in June before it starts decreasing until July (Figure 10). From July to September, the NDVI values show a stable trend, after which the values decrease from September to November. This NDVI pattern shows close correspondence with the monthly rainfall anomalies in the area. However, in the analysis of the relationship of rainfall and NDVI value when $p < 0.05$ is significant, results show that changes in NDVI value versus rainfall is not significant $p = 0.219$, as supported by the lag in one month for NDVI to reach its peak.

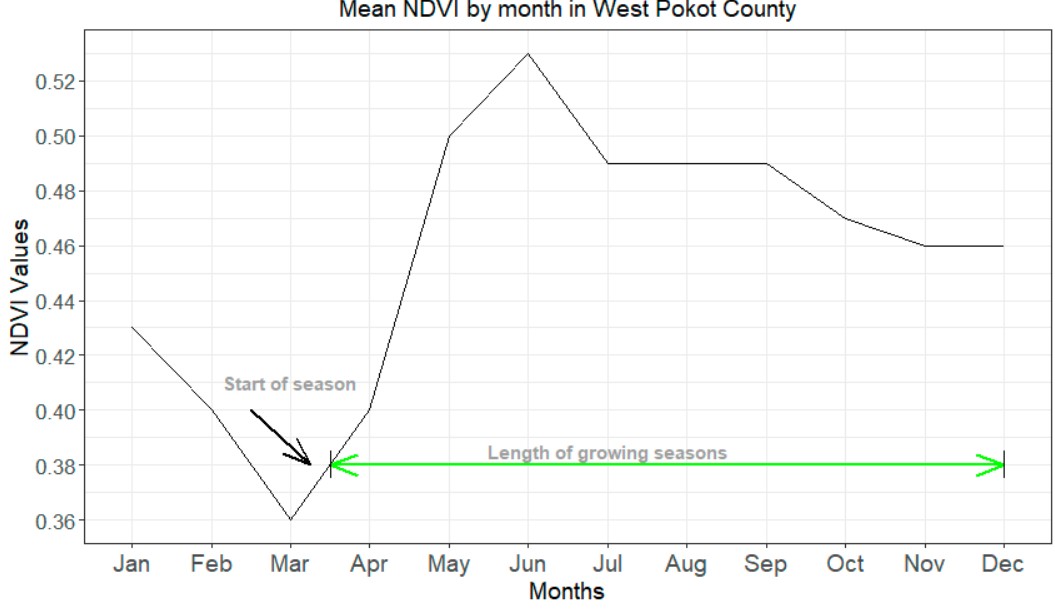

**Figure 10.** Mean NDVI values trend by month in West Pokot.

*3.4. Perception of the Relationship between Food Insecurity and Climate Change and Variability in the Area*

3.4.1. Effect of Climate Variability on Crop Production

Maize and beans were the main crops, while vegetables (such as cabbages and kales), potatoes, bananas and groundnuts were also grown. Respondents reported a high level of food insecurity. A majority (74%) stated they did not have enough food to sustain them previously and at the time of their interviews, and 60% of the respondents projected that food would not be enough in the following year. The respondents reported changed rainfall and temperature variables when discussing food security (Table 4). More respondents in the lower potential lowlands (Kacheliba) reported that they were "greatly affected" by climate changes (75%) while only 14% of the respondents in the high potential highlands (Kapenguria) said food production was "greatly affected". Barely any respondents (3%) noted no effect of climate change, but Kapenguria respondents were the most likely (84%) to report that food production was "slightly affected".

**Table 4.** Extent of climate change effect on food production, as reported in the study areas of West Pokot County (2017, *n* = 124).

| | % Extent of Climate Change Effect on Food Production | | | |
|---|---|---|---|---|
| | **Kapenguria** | **Chepareria** | **Kacheliba** | **All %** |
| Greatly Affected | 14 | 27 | 75 | 34 |
| Slightly Affected | 84 | 66 | 25 | 63 |
| Did Not Affect | 2 | 7 | 0 | 3 |
| | 100 | 100 | 100 | 100 |

### 3.4.2. Change in Crop Growing Seasons

In general, more respondents (44%) reported crop growing seasons were shorter than before, 38% reported no change, and 6% indicated that it varied year to year depending on rainfall distribution (Table 5). Important regional variations can be noted though. While highland (Kapenguria) respondents were more likely to report observing "no change" (55%) than "shorter seasons" (33%), midland (Chepararia) respondents reported "shorter seasons" (46%) more often than "no change" (39%). The lowlands were most affected as compared to the other agroecological zones. In Kacheliba, 57% of respondents reported "shorter seasons" while 25% said it "varied year to year depending on rainfall distribution". A handful of Kacheliba respondents (3%) even reported "longer growing seasons", although this was likely due to their reliance on irrigation from the River Suam for their crop production.

**Table 5.** Reported changes in crop growing seasons in West Pokot County (2017, *n* = 124).

| Temporal Attributes of Growing Season Changes | Kapenguria | Chepareria | Kacheliba | All (%) |
|---|---|---|---|---|
| Shorter | 33 | 46 | 57 | 44 |
| Stayed the same | 55 | 39 | 9 | 38 |
| Longer | 12 | 15 | 6 | 11 |
| Much shorter | 0 | 0 | 0 | 0 |
| Much longer | 0 | 0 | 3 | 1 |
| Varies depending on rainfall occurrence | 0 | 0 | 25 | 6 |

### 3.4.3. Perceived Changes in Crop Yield over Time

The majority of respondents (65%) reported a decline in crop yields (Table 6). Nearly all (97%) respondents in lowland Kacheliba reported a decline, supported by a majority of midland Chepareria respondents (61%), and almost half (49%) of Kapenguria respondents. In the highland and midland, respondents (24% and 27%, respectively) reported steady yields, and 16% of the respondents said crop yields had increased. This could be attributed to those who practiced irrigation in the highlands and some parts of the midlands as an adaptation measure (only 3% in the lowlands reported an increase).

**Table 6.** Reported changes in crop yield over time in West Pokot County (2017, *n* = 124).

| | Agro-ecological zone | | | |
|---|---|---|---|---|
| | Kapenguria | Chepareria | Kacheliba | All % |
| Declined | 49 | 61 | 97 | 65 |
| Steady | 24 | 27 | 0 | 19 |
| Increased | 27 | 12 | 3 | 16 |

### 3.4.4. Weather Forecast Information Access by the Respondents

Respondents reported they had little access to weather forecasts. The most commonly reported source of weather information was from the radio (64% of respondents), ranging from 80% of Kapenguria's respondents to 44% in Kacheliba. Traditional forecasters were the next most often reported source (26%), especially by respondents in Chepareria (34%) and Kacheliba (28%). Other sources of weather information mentioned included TVs, predictions given by local elders, and government agents. Most respondents indicated that they had more confidence in traditional forecasters than the radio (14% of respondents in Kapenguria reported combining radio with traditional forecasts). Even though it was the region with the most problematic agro-ecology, respondents in Kacheliba were the most likely (22%) to report having no access to forecast information, in comparison to Kapenguria (16%) or Chepareria (10%). Overall, respondents reported a desire for forecasts about whether the amount of rainfall would be above average, normal, or below average, distribution of rainfall during the season, and when rains were expected to fall in the area.

### 3.4.5. Measures Taken to Reduce Exposure to the Impacts of Climate Change

The results of the first reduced model are shown in Table 7 and the second reduced model in Table 8.

**Table 7.** Logistic Regression Analysis of 124 respondents on measures to reduce exposure to climate change on food security (Model 1).

| Dependent Variable: Do You Take Any Measures to Reduce Your Exposure to The Impacts of Climate Change on Food Security Risk? | | | |
|---|---|---|---|
| | **Coefficient** | **Std. Error** | **Odds (Exponential of Coefficient)** |
| Intercept | −22.655 | 1311.778 | ~0.00 |
| Gender of respondent (Male) | 1.863 *** | 0.591 | 6.443 |
| Acreage under crop farming | −0.350 ** | 0.160 | 0.705 |
| How has your production been over time since last year up to now? (Remained the same) | −1.345 ** | 0.640 | 0.260 |
| How has your production been over time since last year up to now? (Increased) | 0.736 | 0.804 | 2.087 |
| Do you have enough food for your household currently? (Yes) | 1.205 * | 0.696 | 3.338 |
| Has there been any project targeted at improving the food insecurity condition in your area? (Yes) | −1.680 *** | 0.648 | 0.186 |
| How would you rate the quantity of food in your household? (Bad) | 1.287 | 0.967 | 3.620 |
| How would you rate the quantity of food in your household? (Good) | 2.199 ** | 1.030 | 9.016 |
| How would you rate the quantity of food in your household? (Very good) | 5.430 * | 2.851 | 228.223 |
| Do you depend on relief food when size of land under crop production reduces (Yes) | 2.313 *** | 0.663 | 10.107 |
| How has climate change affected food production for your household since last year up to now? (Greatly affected) | 21.466 | 1311.778 | 2,102,432,864.730 |
| How has climate change affected food production for your household since last year up to now? (Slightly affected) | 20.910 | 1311.777 | 1,205,886,624.298 |
| Observations | | | 124 |
| Log Likelihood | | | −56.623 |
| Akaike Inf. Crit. | | | 139.245 |

Note: * $p < 0.1$; ** $p < 0.05$; *** $p < 0.01$.

**Table 8.** Logistic Regression Analysis of 124 respondents on measures to reduce exposure to climate change on food security (Model 2).

| Dependent Variable: Do You Take Any Measures to Reduce Your Exposure to the Impacts of Climate Change on Food Security Risk? | | | |
|---|---|---|---|
| | Coefficient | Std. Error | Odds (Exponential of Coefficient) |
| Intercept | −0.373 | 0.626 | 0.688 |
| Gender of respondent (Male) | 1.588 *** | 0.591 | 4.892 |
| Acreage under crop farming | −0.163 | 0.125 | 0.850 |
| How has your production been over time since last year up to now? (Remained the same) | −1.157 ** | 0.545 | 0.314 |
| How has your production been over time since last year up to now? (Increased) | 0.479 | 0.700 | 1.615 |
| Do you have enough food for your household currently? (Yes) | 1.493 *** | 0.563 | 4.449 |
| Has there been any project targeted at improving the food insecurity condition in your area? (Yes) | −1.226 ** | 0.531 | 0.293 |
| Do you depend on relief food when size of land under crop production reduces (Yes) | 1.903 *** | 0.551 | 6.708 |
| Observations | | 124 | |
| Log Likelihood | | −64.536 | |
| Akaike Inf. Crit. | | 145.071 | |

Note: ** $p < 0.05$; *** $p < 0.01$.

To achieve a final parsimonious model, we selected all the predictor variables that were significant in the second reduced model as shown in Table 8.

The significant predictor variables of the final model were gender (1 = male, 0 = female), having enough food for household (1 = yes, 0 = no), state of production since the previous year (remained the same, decreased, or increased), any project targeted at improving food security (1 = yes, 0 = no), and being dependent on relief food when size of land under crop reduces (1 = yes, 0 = no). The results showed that:

*Predicted logit of* (*Measures taken to reduce exposure to the impacts of climate change on food security risk*) =
−0.373 + 1.588 * Gender + (−0.163) * Acreage under crop + (−1.157)
* How has been your production over time since last year up to now? [Remained the same] + 0.479
* How has been your production over time since last year up to now? [Increased] + 1.493
* Has enough food + (−1.226) * Food security project + 1.903
* Dependonrelieffood

According to the model, the odds of a male respondent taking measures to reduce exposure to climate change were 4.89 (=$e^{1.588}$) times greater than the odds for a female respondent. Given the same gender, the odds of a respondent with enough food taking these measures were 4.45 (=$e^{1.493}$) times greater than a respondent reporting inadequate food. Holding all other variables constant, a respondent with similar production last year had odds of 0.31 to take up measures, relative to respondents with decreased production. Those with decreased production were more likely to take up these measures to reduce their exposure. Furthermore, respondents who reported a food security related project in the area had odds of 0.29 for taking up measures to reduce exposure to climate change on food security risk. In conclusion, given the same gender and food adequacy state and all other variables holding constant, the odds of a respondent that was dependent on relief food when size of land under crop reduces were 6.71 (=$e^{1.903}$) times greater than a respondent who had never received relief food. The model AIC was 149.64.

## 4. Discussion

### 4.1. Characterizing Annual and Seasonal Rainfall and Temperature from 1980–2011 in West Pokot County

The increase in mean temperature confirms climate change and variability in West Pokot County in excess of the global mean temperature increase (+0.85 °C between 1980 and 2012) shown by IPCC [20]. The increase in temperature over time in both the highlands and lowlands of West Pokot agrees with Naanyu [22], who anticipated warmer temperatures in Kenya. Collins [60] indicated that rapid warming from 1979 onwards was witnessed in Kenya. Ogutu et al. [61] focusing on 21 arid and semi-arid (ASALs) counties of Kenya indicated warming of these counties. In Africa, Issahaku et al. [62] and Safari [63] recorded similar results of significant warming temperature trends in Ghana and Rwanda. In addition, King'uyu et al. [64] found evidence of significant warming in the East Africa region.

Kurukulasuriya [65] warned that high temperatures are harmful to productivity, confirming that global warming is likely to have devastating effects on agriculture unless farmers take adaptation measures to counter the impact of climate change. Rising temperature would expose millions of people to drought and hunger. Warming leads to higher rates of evaporation [66], puts additional stresses on water resources [67], and causes sterility during the reproductive period [68]. These effects on fertilization and grain formation resultin diminished yields [69] and escalated livelihood vulnerability.

With rainfall being a prime input and requirement for plant life in rain-fed agriculture, the occurrence of dry spells has particular relevance to rain-fed agricultural productivity [70]. There is a highly variable trend in rainfall in West Pokot, with the number of years recording rainfall of below-average outnumbering those of above. Akponikpe et al. [71] also reported similar trends of high variability (a coefficient of variation of 57%) in temporal annual rainfall in the Sahel region, while Mzezewa et al. [72] reported even higher coefficients of variation for seasonal (315%) and annual (50–114%) rainfall at a semiarid site in Limpopo Province, South Africa.

However, several studies show varying results for rainfall trends in Kenya. For example, in the Western part of Kenya, Githui et al. [73] reported increasing amounts of annual rainfall (+2.4 to +23.2%) from the 1970s up to 2008, while in the Eastern part of Kenya, Rao et al. [74] found no discernible increasing or decreasing trend either in annual or seasonal rainfall. However, Watson et al. [75] recognised that rainfall results around the region are not consistent. Christensen et al. [76] found a discrepancy in observed rainfall tendencies from the 4th IPCC assessment, particularly relating to the long rains (June–August) precipitation, noting that there is a substantial spread between the rainfall projections of different models. The IPCC report does acknowledge that the models it used often have difficulty representing the key processes affecting rainfall and its seasonal distribution in Eastern Africa, with typical correlations of less than 0.3 over land surfaces [77]. Lyon and Vigaud [78] on the other hand argued that the models may be accurate but extreme weather events such as El Niño could cloud the overall picture while explaining the 'East African paradox'.

### 4.2. Land Use Land Cover (LULC) Changes

The changes in LULC illustrated both the impacts of irregular rainfall over the period from 1980–2011 and the complex, human responses to this variability. As noted above, low rainfall years in the 1980s and 1990s contributed immensely to reduced green cover, especially around the highland regions. The gradual decrease in rainfall and longer dry spells resulted in expanded cropland area, which was viewed as a quick coping strategy to the reduced yields that followed failed rains. Farmers began opening croplands in the lowlands, especially along River Suam. Although the changes in LULC do not directly imply a degradation of the land, improper handling of land use may lead to a "landslide effect" resulting in new problems [79], including exacerbating land degradation and soil erosion, which might reduce the overall yield of food crops. The slight increase in forest cover in 2010 reflected renewed government efforts in afforestation [23], although trees

take time to mature. The annual temperature trend with more gradual peaks (1983, 1992, 1995 and 2009) preceded rainfall seasons. This may be explained by the effect of reduced vegetation cover and the expansion of land uses that expose the land such as croplands and settlements. Fall et al., [80] indicated that climate forcing from LULC dynamics significantly impacts local temperature trends.

However, changes in cropland and forestland can be explained not only by reduced primary productivity but also the pressure that the county's dependence on rain-fed agriculture has put on forest cover, which was overutilised, and converted to agricultural lands, especially around in the highlands [44,81]. This makes agricultural expansion the dominant direct cause of deforestation in the area and agrees with the County Government of West Pokot [82], that encroachment and clearing for cultivation, demand for timber and fuel wood has accelerated deforestation. Similarly, Brink and Eva [83] reported a massive increase of agricultural land in East African drylands over the period 1975–2000 by at least 123,413 ha at the expense of forests (71,325 ha) and natural grassland vegetation (58,894 ha). A study done by Wasige et al. [84], noted LULC changes in Kagera Basin, spanning across Burundi, Rwanda, Uganda, and Tanzania between 1901 and 2010. Their study indicated a decline in natural savanna grassland by 15.4% between 1901 and 2010 while cultivated land increased considerably. Amsalu et al. [85] and Gessesse [86], warned that land use change is brutal, with agricultural land areas in Ethiopia expanding at the expense of natural vegetation cover into marginal areas without any appropriate conservation measures. A similar negative feedback can be seen in our study, where grassland area increased following years of heavy rains, leading to increased livestock keeping, which in turn resulted in overgrazing and decreased woody cover.

The increased open water area between 2000 and 2010 reflects improved hydrology and water storage afforded by farmers' increased water harvesting and reafforestation measures. This agrees with studies conducted by Du et al. [87] who reported that water bodies increased from 125,369 to 1,315,574 ha, at an annual growth rate of 12,389 hectares per year in Jiangsu Province of China during 2000–2005. Settlements have increased in the County following population increase and urbanization, stimulated in part by national decentralization initiatives [44]. The population of West Pokot County (which was a District until 2010) has increased drastically but steadily, from 158,652 in the 1979 census, to 225,449 in 1989, 308,086 in 1999, and 512,690 in 2009 [88,89]; a net increase of 223% over the 30-year period, or +7.44% per annum.

### 4.3. The Phenology of Agricultural Vegetation in West Pokot County

The consistent increase in vegetation greenness in the NDVI time series in West Pokot County corresponds with the rainfall patterns of the seasonal calendar [90,91]. Pricope et al., [92] also stated that in the horn of Africa, the spatial distribution of NDVI is directly related to precipitation and terrain. In a similar study conducted in the Turkana area, Omondi [93] found that the peaks and lows of the vegetation greenness naturally represented the rainy and dry periods and that there was a good correlation occurred between average rainfall and NDVI for monthly data with a trend of increasing NDVI with rainfall. Omondi added that NDVI has stronger linkages with monthly rainfall anomalies than any other climatic variable. However, a study conducted by Regmi et al. [94] in Nepal did not find a significant correlation between NDVI, temperature, and precipitation. Topographical effects may affect the correlation coefficient between NDVI and climatic parameters [95].The author indicates that caution is required when assessing correlation with NDVI in such specific locations.

The low NDVI values for West Pokot from January to February partly reflected the harvesting and land preparation that takes place in these months. The NDVI peak in June corresponded with the maturing of crops planted during the long rains (from March to May) to grow with sufficient soil moisture. The decrease in NDVI values from June to July followed the reduction in greenness as these same crops were harvested. Land preparation and planting begin anew in August until September as the short rains begin, hence the

stable but lower NDVI values reflecting bare land and plants which have not germinated from the ground. Harvesting of these second crops started in November, which accounted for decreasing NDVI and greenness. This means that prolonged dry-spells during cropping seasons directly impact the performance of crop production [96]. Alternating dry-spells occurring and exceeding the same durations show the high risks and vulnerability that rain-fed smallholder farmers are exposed to in the study area [96]. Often, prolonged dry-spells are accompanied by poor distribution and low soil moisture for plant growth during the growing season.

Although April–May and August were the peak rainfall months in the study locations, June and September were the peak NDVI months, meaning there was a roughly one-month lag period. Previous studies by Anyamba et al. [97] reported a 1–3 months lagged response of rainfall and NDVI in Eastern Africa after the 1997/ 1998 El Niño event, 1 month in West Africa, and 1.5 months in Southern Africa [98]. The lag showed that vegetation does not respond directly to rainfall, but rather to soil moisture, which is a multi-month integral of rainfall [99].

*4.4. Perceptions of the Relationships between Food Insecurity and Climate Change and Variability in West Pokot County*

4.4.1. Perception of climate change and climate variability in West Pokot County

Delayed onset and early cessation of rainfall and increased temperatures were variables more recognized by respondents than the total rainfall amount in a season. This perception conforms with empirical evidence from agronomic studies [100,101]. There was more consistency among observations related to temperature increase and climate data.

Respondents agreed most about an apparent decrease in the amount of rainfall. These results agree with several studies including those of [102–104] where communities in Mexico, East Africa, and India, perceived a decrease in rainfall amount and duration. Other studies [105–107] found that more than 50% of the respondents asserted rains were decreasing. Even though our findings agree with this other research, the local perception of a decreasing rainfall trend was not supported by meteorological data. The dataset shows no discernible trend, suggesting that respondents were instead influenced by variability in rainfall between years and extreme events. This could also be attributed to their observations of crop stress, dry fields, and drought causing them to link such observations with perceived reductions in rainfall [108], or with their recent experiences of flood/drought/poor rain associated with increased climate variability [109–111].

Similarly, Mulenga and Wineman [112] noted a clear overlap between farmers' observations and patterns found in the meteorological records. Slegers [113] found that farmers in Africa hold a definition of drought that is broader than a simple lack of rain. Rather, they focus on the aggregated impact of multiple climate variables. Farmers' memory of past events can be faulty as well as their failure to differentiate between climate (the statistical expectation) and weather (what we get) patterns [114]. However, a study by Tierney et al. [115] supported the findings of the recent decrease in rainfall over the Greater Horn of Africa (GHA), especially during the long rains. Liebmann et al. [103] reported a decrease in rainfall during the period of 1979–2005 over East Africa. Analytical results for Kenya indicate that the precipitation in Kenya is not uniformly distributed through time and space [116].

4.4.2. Perception of Impacts of Climate Change and Variability on Crop Productivity

Respondents perceived that temperature increase and rainfall variations reduced crop production, particularly of maize, which is a staple in the area. This agrees with other studies [31,117–119] in SSA showing evidence of negative climate change impact on crop yield including maize, a major staple cereal food crop, with huge implications for the arid and semi-arid regions [120]. This might, however, relate to the changes in management rather than climate change, such as the increasing efforts to grow maize in larger areas and in more marginal environments with no or insufficient new inputs to

maintain it. Nevertheless, some of the studies [121,122] have acknowledged the interplay of other non-climatic factors as well. Kangalawe et al. [122] stressed the need to quantify the magnitude of impacts of climate change and isolate them from non-climatic factors with compounding effects. They noted that it may not be possible to separate the impacts of climatic and non-climatic factors on crop production and agriculture entirely as they interact and intertwine with each other generating impact.

Impacts on crop productivity were more notable in the lowlands and some parts of the midlands than in the higher potential highlands. This could be associated with the recent, repeated drought events restricting crop production or could be linked to various environmental changes (temperature increase, rainfall irregularity, degradation of soil structure, etc.) that reduced water availability and agricultural yield in the dry lowland areas. Kalungu and Harris [123], in their study of climate variability in the semi-arid and sub-humid regions of Kenya showed that 74.4% of farmers in the semi-arid region perceived changes in crop productivity for the past 10 years against 57.85% of farmers from the sub-humid region. IPCC [2] had indicated that crop productivity in low tropical and dry areas is projected to decrease with an increase in temperature of 1 °C to 2 °C. Even though Fischer and van Velthuizen [124] indicated that the overall impacts of climate change on food production will have a positive impact on food, results will vary by region. Jones and Thorton [125] showed that maize production in Africa and Latin America would reduce by 10% by 2055 and recommended that climate change impacts be assessed at the household level so that the poor who depend on agriculture can be targeted for advice.

Rainfall variability, particularly during the short rains season, is the major constraint noted in West Pokot. Unreliable short rain harvests mean many households now rely exclusively on the harvests from a single, long rains crop production each year. Ngigi [126] also noted that most areas characterized by low and erratic rainfall, concentrated in one or two rainy seasons may result in high risk of droughts, intra- and off-seasonal dry spells, and frequent food insecurity.

### 4.4.3. Perception of Weather Forecast Information

The respondents practiced their farming with minimal knowledge of the ongoing climate changes and possible impacts on seasonal weather, meaning they may not be adequately empowered to respond and adapt to the projected magnitude of these changes. Gwimbi's [127] study in the Gokwe District of Zimbabwe found that more than 70 percent of the surveyed farmers lacked access to timely weather forecasts. Ziervogel et al. [128] and Lemos and Dilling [129] highlighted that forecasts had not been extensively embraced and their effective utilization has lagged, particularly in marginal areas. The range of information sources in West Pokot was extremely limited, unlike other studies conducted in Bangladesh, Ghana, and Uganda by Chaudhury et al. [130], which found diverse information sources such as radios, newspapers, mobile phones, public announcements at schools and during religious gatherings, and print media as important channels. Cherotich, et al. [131] argued that the choice of the dissemination channels could influence access and use of climate information and service disseminated enable the vulnerable groups exposed to climatic hazards to build adequate response capacities. Education has contributed to imparting awareness among people [132].

Hansen et al. [133] reported that radio and ICT-based communication offer immense potential to support the delivery of climate information support services, but cannot replace the trust, visual communication of location-specific information, feedback, and mutual learning that face-to-face interaction provides. The use of ICT in West Pokot was however very low. Ballantyne, Labelle, and Rudgard [134] contend that the use of ICTs in rural areas is limited by lack of awareness, skills, training, and a shortage of capital resources for sustainability. Our respondents had more confidence in traditional forecasters than radio because to them, these predictions are location-specific. Indeed, even existing scientific data have issues accurately predicting some parameters, such as the duration and coverage

of drought [135], which in some places is aggravated by the fact that the forecasts are not location-specific [136].

Other studies have argued that some farmers who prepare their land and plant their crops based on traditional prediction techniques are forced to replant them due to an unexpected dry spell after the early rains [137,138]. On one hand, some scholars feel skeptical about the accuracy and reliability of traditional prediction methods under current weather and climate change and variability [139,140]. Others [141,142] acknowledged and emphasized the importance and use of local knowledge for weather and climate prediction, decision making, climate change adaptation [143,144], and to complement scientific information [140,145,146].

*4.5. Measures Taken to Reduce Climate Change Impacts on Food Security*

Apart from receiving relief food, the people of West Pokot are employing other coping mechanisms, however men took more measures than women. This could be attributed to patriarchal land ownership rights in the area affecting adaptation decisions, and agrees with [147] that different socio-economic, environmental, and institutional factors affect the climate change perception and adaptive behavior of farmers. Nelson et al., [148] notes that existing gender imbalances in agriculture mean that women are potentially at a comparative disadvantage in terms of participating in and benefiting from site-specific climate actions on the ground. As observed in other studies in the Upper West Region of Ghana, female farmers showed preference for adaptation measures that have benefits that could be realized in the short-term because of the constraints they faced in accessing productive resources such as land and labor [149,150]. This suggests the need for incorporating gender-based assessment of climate change adaptation in planning for adaptation interventions [151].

Dependence on relief food is highly regarded as a coping strategy by the Pokots. In the years characterized by prolonged drought and famine, many Pokot families relied on famine relief for survival. For example, in 1980 and 1981 which had minimal rains [152] and the 1984/85 period [153], West Pokot benefited from famine relief and food aid. In 2013, the county suffered massive flooding that submerged untold hectares of maize, millet, and sorghum, and was therefore, supplied with relief food [154]. In 2015 and 2016, 600,000 residents faced acute hunger as the maize crops withered and livestock starved under extreme heat and lack of rainfall in the long rain season [154]. In 2017, over 40,000 residents were in dire need of water and food in West Pokot [155].

Drought mitigation strategies including relief are deemed to improve socio-economic conditions, reduce pressure on land (crop and grazing), and reduce vulnerability [156]. The Pokots could opt for relief food due to inadequate production to feed the population and a lack of adequate coping mechanisms [153]. However, this is a short-term solution and increases the level of dependence in the long-term [153], without protecting them against a recurrence of hunger or enabling greater self-sufficiency [157].

**5. Conclusions**

The results confirm that the climate in West Pokot has changed and is already having implications for food security. Semi-arid regions are inherently dynamic, with high degrees of interannual and spatial variability, but increased uncertainty and variability amplify the vulnerabilities of existing farming systems. While respondents might attribute perceived changes in local vegetation to climate changes, the GIS and rainfall data suggest that human-induced conversion is more likely responsible for converting grassland/forestland to cropland. As these conversions interact with ongoing climate change and variability, the competing land uses involving forestland, cropland, settlements, wetlands, and grassland, are likely driving the system toward a less viable use of the environment. NDVI correlates with rainfall received, with a one-month lag, and can be useful in monitoring and managing drought on a near real-time basis and in trend analysis. The observed NDVI trends in West Pokot, however, cannot be exclusively explained by rainfall anomalies, since there are other human factors that impact vegetation dynamics including LULC change.

The local populations are responding to the perceived changes in diverse ways, which means strategies for adaptation cannot be generalized and should be more site-specific. Policymakers and development agencies should focus on formulating and implementing policies and programs that minimize overreliance on relief food and promote farm-level adaptation strategies such as agroforestry, reforestation, and climate smart agriculture where drought-resistant trees and crops could be introduced or encouraged according to the agroecological zone. Trade-offs between increasing cropland and the subsequent reduction in grassland as a coping strategy by the Pokots need be investigated in terms of socio-economic and ecological sustainability, and their effects on other ecosystem services. GIS and climate data can be used to complement the existing local knowledge.

Better quality, timely, and site-specific scientific weather forecasts could help bolster local knowledge systems and adaptation practices. This study showed that at present, West Pokot people have little access to, and make minimal use of, weather information, and deem its present value quite low. Increasing the availability of weather stations at the local level and enhancing the capacity to collect and analyze weather information could enhance appropriate adaptation strategies. Farmers will appreciate timely information on the amount and distribution of rainfall and the expected time of onset and cessation during cropping seasons. The government should support continuous climate change monitoring, intensified early warning systems, and the dissemination of relevant information to farmers. Further research could focus on investigating climate variables and human-induced factors in vegetation variability, as well as the long-term monitoring of the arid ecosystems.

**Author Contributions:** Conceptualization, E.B.O., J.J.R.; methodology, E.B.O., J.J.R., L.D.; formal analysis, E.B.O., L.O.; writing—original draft preparation, E.B.O., J.J.R., L.D.; writing—review and editing, E.B.O., J.J.R., L.D., L.O. All authors have read and agreed to the published version of the manuscript.

**Funding:** The field research was funded by the Canadian Social Sciences and Humanities Research Council (SSHRC), under the grant # 410-2011-2323 (*"Global Climate Change and Kenya: Vulnerability and Adaptation of Livelihoods Under Environmental Stresses (VALUES)"*).

**Institutional Review Board Statement:** The study was approved by the Research Ethics Board of the University of Ottawa (file 05-12-19B).

**Informed Consent Statement:** Informed consent was obtained from all subjects involved in the study.

**Data Availability Statement:** Data is available upon request from the lead author.

**Acknowledgments:** This research was made possible by the kindness, patience, and hospitality of the people of West Pokot. Thanks also to Obade, Koske, and Kung'u of Kenyatta University for their honest guidance and constructive criticism of earlier versions of this research, and to four anonymous reviewers who improved earlier drafts of this paper. Despite the good efforts of these many people, the authors must remain accountable for any deficiencies in the final product.

**Conflicts of Interest:** The authors declare no conflict of interest.

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
