# Peer review of "The Relationship between Climate Change, Variability, and Food Security: Understanding the Impacts and Building Resilient Food Systems in West Pokot County, Kenya"

_sustainability, doi:10.3390/su14020765_

Round 1

Reviewer 1 Report

There are many grammatical errors. The authors are using present tenses in reporting a study that they already conducted.

The objectives of the study are not given

Results interpretation is very poor and weak. There are sections with a mix up of results interpretation and discussions yet the authors have an independent section on discussion.

Discussions are very weak, insufficient, disconnected, lacks coherence and not convincing.

The conclusion section needs to re-written because it is more of a summary of the findings instead of an answer to the study objectives/hypotheses (which are not outlined)

Page 1

Line 15; add ‘observed’ climate data

Line 16 It is Normalized Difference Vegetation Index

Line 16 use long-term not longer

Line 18: Delete has become more variable and use …that there is climate variation.

Line 18 Delete A large and use The majority of the respondents

Line 18 Delete also

Line 19 Delete warming

Line 19 why more likely to report? Yet you are supposed to be reporting the results

Line 22 Temperature data

Line 25 showed

Line 40 “are of exceptionally high vulnerability’ lacks connection with the rest of the statement

The first paragraph has very long sentences, which tires a reader and loose meaning. The authors should aim to make them concise

Page 2

Line 60 -64 Too much verbosity for one sentence. Be concise

Line 70 active and healthy life..

Line 70 Delete… ‘for all household members”

Line 74 ..should aim to achieve….

Line 79 Delete Most

Line 79 ..conducted in Kenya not on Kenya

Line 83 Delete ..addresses this gap and

Line 86 use ..in designing not for

Line 87 Delete for

Page 3

Line 99 Outline the specific objectives and hypotheses of the study.

Line 116 Besides having a table describing the study area, I suggest you also draw a map illustrating the descriptions outlined in the site selection section

Line 120 Describe the stages in the multistage sampling technique so as we have a clear understanding how you narrowed to 124 respondents

Line 136 …..and bands 1, 2, 3…

Line 138 Delete also

Line 138 Which is that “more information”?

Page 4

Line 168 – 172 Delete. It is redundancy

Line 177 – 178 Rainfall being unpredictable and unreliable does not confirm that it is not sufficient water long enough for crop cultivation. It should be about amount and coincidence with the cropping seasons

Page 5

Line 183 there should be a space between numbers and units for example 1347.9 mm

Line 189 Delete Driest

Line 192 & 193 Move ..”such as maize and beans” next to crops …….

Line 195.. a respondent lamented..was this by a key informant? If yes, be specific, If no, was it a question in the interview schedules? Then it should be statistically analysed

Line 195 -196 ..some recovery..this street English. Use better terms

Page 6

Figure 2 The seasonal series doesn’t seem to show any variations

Line 202 -203 Cite the table or figure showing the said findings

Page 7

Line 214 Separate numbers and units

Line 223 Separate numbers and units

Write the unit (℃) correctly in the whole paragraph

Line 227 showed

Line 228 Use revealed and Delete those

Page 8

Line 236 showed

Line 242 showed

Line 243 ..revealed

Page 15

Line 277 – 281 Cite the figures or table that illustrates this information

Line 282 Replace results were not quick with “rains were not sufficient”

Line 281-285 is more discussion not results interpretation

Line 289-293 is more discussion not results interpretation

Page 21

Line 340 – 344 Separate numbers and units

Line 340 Define “normal” vegetation and “poor” vegetation

Line 361 – 370 I suggest you re-write the paragraph in reporting tense and avoid doing discussions.

Also cite the figure in text. This applies to most of your results interpretation sections, the lack tables or figure citations

Page 22

Line 283 Delete “Very large majorities indicated that rainfall is declining..and Use The majority indicated the rainfall has been declining

Line 382 – 391 Which table or figure illustrates those results

Line 388 Instead of using section 3.1.1 cite the figure or table

Line 396 ..“the time of their interviews”..how about previously? You cannot ascertain food insecurity within the few days you did interviews. It has to be for a good period

Line 397 Delete considered and use reported

Other comments are highlighted in the pdf document

Reviewer 2 Report

Thank you for giving me opportunity to review this intriguing article. I have read this article with great interest which deals with assessment of  the impacts of climate change and variability on food security in Kenya from 1980-2012 by exploring local knowledge and perceptions and combining them climate data (rainfall and temperature), seasonal vegetation change. Over all the paper is well written and is relevant to the journal’s theme. I recommend the publication of this article in the journal as it could be interest of journal’s audience. However, I suggest the authors to incorporate the following suggestions.

Introduction.

Discuss the relevance of your study in regard to other regions, lets say Asia, what is the extent of available literature in vulnerable regions such as south asia.

about 223 million people are currently undernourished in SSA, please name the acronym/abbreviation at the first mention, what is SSA.

he situation has worsened and the frequency of the country’s famine cycles has 66 reduced from 20 years (1964-1984) to 12 years (1984-1996), to 2 years (2004-2006) and cur- 67 rently to annually (2007/2008/2009/2010/2011/2012) [23]. So what are you implying, the famine occurs ever year?

Line 76, please correct typos

Data and Methods

2.1 Site selection

Please mention the source of information while indicating precipitation in the selected three categories of the region. Add relevant citation. Besides, what are the temperature differences, if any, among the selected three categories of the region, its better to indicate along with precipitation records.

Questionnaires were used to elicit information for ground truthing, verification, 118 providing basic information and for perception study

What type of questionnaire was used, was it a structured or semi-structured, and mentioned If you did a pre-test.

20 individuals from various organizations were interviewed as key inform- 122

Please mention what organization did you consider for KII, it is important to know whether they were relevant institutions, consider this study [1].

Data analysis

While the authors comprehensively discussed the used models for satellite data, but regarding the analysis of survey data, the section needs improvement. The authors mentioned they used Multiple regression model for the adopted adaptation measures. I suggest adding more information why this model was used, as some studies used alternative models, for example [2] used binary regression, while [3] employed multivariate regression. Please consider these studies to improve the justification rationale of selecting the used model.  And do mention for which year/years did you ask the adopted measures from the farmers, was it for about the previous year or the current year.

Please also mention the time of survey for better understanding of the readers.

  1. Results
    3.1.1 Rainfall trend analysis

While comparing actual climate series data with farmers’ perception you need to include relevant citation to compare your results. This aspects critically lack at various instances. Please improve, consider the suggested studies to improve this part, or pick previous researchers in Kenya to improve your discussion.

Again you said,

Respondents did, however, believe 202 that rainfall was diminishing over time, which is not supported by these data.

Why the trend could align with farmers perceptions, add details/explanation

Same applies for the following subsections of this section.

The results obtained from the imagery and presented as figures seems interesting while authors have tried to cover major aspects and changes, the conclusion, discussion, and comparison with the studies already done severely lacks throughout this section

3.4.1 Respondents’ views on climate variability 1980-2012

How did you assess perception, did you use likert scale, if so add information in here or in the method section, or simple Yes/No, questions?

Besides, how about the authenticity of the perception, as you mentioned that the perception were collected for years 1986-2010, which is quite long period while asking farmers/people. There might be younger farmers who may not be born yet, so it means reader need information on farmers/respondents’ personal traits age, education etc. please see the articles I suggested

Same applies for the following subsection of this section.

Most 439 respondents indicated that they had more confidence in traditional forecasters than the 440 radio.. what is it so, while other studies [4, 5] show that farmer now use ICTs to access weather information and it is quite useful for them, why is it different in case of Kenya? Need explanation and discussion/comparison

Authors have well explained the results from regression models, just a few citation needs to be there, which I rarely saw in that section too.

  1. Conclusions

The section is well written,  no comments

Khan, N.A., et al., Formal institutions' role in managing catastrophic risks in agriculture in Pakistan: Implications for effective risk governance. International Journal of Disaster Risk Reduction, 2021: p. 102644.

Reviewer 3 Report

This manuscript describes a study that sought to evaluate the impacts of climate change on food security in Kenya. The subject matter fits with the scope of the journal.

Generally, the article is reasonably well written, clear and unambiguous. Referencing is food and the Introduction provides an adequate summary of the relevant background science. However, a strong rationale for the study is not evident. It is well established knowledge that climate change will and, indeed is impacting on food security globally but, perhaps, more significantly in the developing world. What was the driving research question for this study? What gaps in science did it seek to fill that could not be filled by current knowledge or by extrapolating existing knowledge? Who benefits and how? The text refers to ‘…systematic data of such relationships…’ please clarify this statement and explain why the data is important in this context and how it will be used and who by. Some information is given but this is not sufficiently clear.

The section of Data Acquisition requires more detailed information. The text states that 124 households were selected via a multistage sampling technique. Please expand providing information on selection criteria and evidence that 124 households is statistically valid for the study area and study context. This part of the article also refers to 20 individuals who were interviewed. How were these individuals selected, what was the purpose, what format did the interview take etc.? What is the value to this particular study of knowing that the survey respondents do not have little access to weather forecasts?

The Methodological section does need much greater clarity. How exactly as the survey and questionnaire data used/analysed?

The analytical side of the study is not new science. Whilst interesting it is just a re-iteration of pre-existing data and not new science. Considering the text given in Section 3.4.1 it is not clear why a questionnaire was needed to confirm what actual hard data states. It is also unclear to me what the data gathered by the surveys and questionnaires gives beyond the hard data.

My main concern with this manuscript is that it does not contain much if any new science. At Line 619 it states that it is clear that climate in the region has changed and this is impacting on food security – did we not know that previously?

Other issues

  • Please spell all abbreviations out in full the first time they are used in the main test. e.g. Line 52 FAO, Line 58 SSA, Line 89 LULCC, Line 95 NDVI etc.
  • Line 149 ‘…carefully…’ is a very subjective term. Who decided what this meant!

Reviewer 4 Report

I would like to thank the authors for the efforts made to produce this manuscript. I only have two comments, one on the research idea and one on the methodology.

  1. It is worth noting that research on climate change impacts on agriculture is abundant. We need more research on adaptation and resilience. Therefore, this drives me to reduce my vote on originality.
  2. In the methodology, the authors state the number of households surveyed, without giving the total number of households in the three study areas concerned. What if the selected number is not representative? The authors should argument this choice.

Round 2

Reviewer 1 Report

The article has improved but the English language needs some improvement.

There is a mix up of conclusion and recommendations. First, give a concise conclusion of the findings then do recommendations. For example in the first paragraph Line 1234-1240 Those are recommendations and not conclusion

Line 1275 The study has shown…Use past tense..The study showed

Reviewer 3 Report

This manuscript describes a study that sought to evaluate the impacts of climate change on food security in Kenya. The subject matter fits with the scope of the journal. This is the second time I have reviewed this manuscript.

I am pleased to see that the authors have addressed my concerns previously communicated. I still do not think this study offers much in terms of filling gaps in knowledge but I have no concerns which I believe should stop its publication.
